# Effect of mobile food environments on fast food visits

Bernardo García Bulle Bueno[1], Abigail L. Horn[2,3,9], Brooke M. Bell [2,4], Mohsen Bahrami [1], Burçin Bozkaya [5], Alex Pentland[1], Kayla de la Haye[6] & Esteban Moro [1,7,8] ✉

Poor diets are a leading cause of morbidity and mortality. Exposure to low-quality food environments saturated with fast food outlets is hypothesized to negatively impact diet. However, food environment research has predominantly focused on static food environments around home neighborhoods and generated mixed findings. In this work, we leverage population-scale mobility data in the U.S. to examine 62M people's visits to food outlets and evaluate how food choice is influenced by the food environments people are exposed to as they move through their daily routines. We find that a 10% increase in exposure to fast food outlets in mobile environments increases individuals' odds of visitation by 20%. Using our results, we simulate multiple policy strategies for intervening on food environments to reduce fast-food outlet visits. This analysis suggests that optimal interventions are informed by spatial, temporal, and behavioral features and could have 2x to 4x larger effect than traditional interventions focused on home food environments.

Poor diets, including the over-consumption of foods that are energy-dense but nutrient-poor, that have excess sugar and/or sodium, and that are ultra-processed, are a major cause of diet-related disease and mortality[1–3]. Poor diets led to 11 million deaths globally in 2017 (more than tobacco)[4], largely due to their causal role in major chronic diseases, including obesity, type 2 diabetes, some cancers, and heart disease[5]. Exposure to, or spending time in, certain built food environments is hypothesized to impact diet and related diseases[6,7]. Low-quality built food environments are generally categorized into two types. Food deserts are defined as areas with low access to healthy foods (e.g., neighborhoods where a majority of residents live more than 0.5 or 1 mile from a supermarket, a key source of affordable, healthy food)[8]. Food swamps are areas saturated with food outlets selling unhealthy foods, often defined as neighborhoods that have a

higher number of fast food outlets (FFO) and convenience stores, or a high ratio of these outlets relative to healthier food outlets[9,10]. Both of these types of low-quality food environments are frequently concentrated among low-income communities and communities of color in the US and contribute to inequities in nutritional health[11,12]. It is hypothesized that exposure to food swamps can nudge people to consume unhealthy food (e.g., fast food) due to a cost decrease via lower food prices or less time needed for transactions[13], or through structural or social cues to behavior[14]. In contrast, food deserts are hypothesized to create barriers to accessing affordable healthy foods, which can lead people to make less healthy food choices that they would otherwise avoid.

To date, research into the relationship between food swamps or deserts and food choice has predominantly focused on predefined

[1]Institute for Data, Systems, and Society, Massachusetts Institute of Technology, Cambridge, MA 02139, USA. [2]Department of Population and Public Health Sciences, Keck School of Medicine, University of Southern California, Los Angeles, CA 90089, USA. [3]Department of Industrial and Systems Engineering, Viterbi School of Engineering, University of Southern California, Los Angeles, CA 90089, USA. [4]Department of Chronic Disease Epidemiology, Yale School of Public Health, Yale University, New Haven, CT 06510, USA. [5]Sabanci Business School, Sabanci University, 34956 Tuzla, Istanbul, Turkey. [6]Center for Economic and Social Research, University of Southern California, Los Angeles, CA 90089, USA. [7]Grupo Interdisciplinar de Sistemas Complejos (GISC), Department of Mathematics and GISC, Universidad Carlos III de Madrid, 28911 Leganés, Spain. [8]Network Science Institute, Northeastern University, Boston, MA 02115, USA. [9]Present address: Information Sciences Institute, Viterbi School of Engineering, University of Southern California, Los Angeles, CA 90292, USA. ✉e-mail: esteban.moroegido@gmail.com

local and static food environments[15,16], largely of the neighborhood around the home, with schools and workplaces to a lesser extent. While exposure to both types of food environment has been associated with increases in unhealthy eating and diet-related disease, overall, findings are mixed and predominantly null[17–20]. Furthermore, most of these studies have been cross-sectional and fail to establish a causal relationship between neighborhood food environments and unhealthy diets. Despite this limited evidence, there has been considerable interest from federal and local policymakers and private funders in supporting policy interventions to improve neighborhood food environments. These include investments by the U.S. Healthy Food Financing Initiative of $270 million plus $1 billion in leveraged financing to support healthy food retail in underserved neighborhoods since 2010[21], and 'fast food bans' implemented in select Los Angeles neighborhoods by the city council using zoning regulation to restrict the opening of new FFO.[22] Across numerous evaluations, these interventions have demonstrated no meaningful impact on diet quality or diet-related disease outcomes[22–27]. A better understanding of the relationship between food environment exposure and use, diet, and diet-related disease will be critical to designing more effective interventions to food environments.

The limited focus on residential and static food environments may be one explanation for these mixed results, given that a growing proportion of food acquisition and consumption occurs miles from our homes. For instance, Cooksey et al.[9] presented findings that food swamps predict higher rates of obesity at the neighborhood level. However, their results are weaker in neighborhoods where residents are more mobile (i.e., more residents who travel to work by car or public transport). Among US residents, food away from home (vs. foods prepared at home)–the vast majority coming from fast food and full-service restaurants–constitutes one-third of total energy intake, and one-half of food budgets[28]. Thus, a major source of exposure to and use of food environments is unlikely to be captured by existing research foci and methods. Additionally, these studies often test whether exposure to food swamps or deserts predicts nutritional health, without incorporating information on the food outlets that individuals actually visit. Given the well-documented biases of survey data to capture detailed human movement and dietary intake[29], small studies (often <100 people) have begun to use tracking technologies to map how people move through their environment to acquire food over brief periods of time (e.g., one week)[30–34]. However, this has not been studied at scales large enough to capture habitual patterns of food environment exposure over extended time intervals or statistically significant effects of those food environments on peoples' behavior. Overall, a major gap in the literature is detailed evidence of the food environments people are exposed to as they move around, both at and beyond where they sleep and work (i.e., mobile food environments), the food outlets they actually visit in these environments, and causal designs capable of investigating how mobile food environments influence diets and diet-related disease.

In this study, we use a large, privacy-preserving, population-scale mobility dataset spanning a 6-month period during 2016–2017 and 11 metropolitan areas in the US to examine peoples' visits to food outlets (FO) and FFO in and beyond their home neighborhood and to investigate how these FFO visits are linked to features of the mobile food environments they are exposed to throughout their daily routines. Mobility data allow us to observe a diverse and heterogeneous population body[35,36]. They allow us to observe when and where FFO visits happen among this large, diverse population, and thus to understand the individual- and environment-level variables that condition that decision over other food choice alternatives. Moreover, they allow us to find structural, randomized shocks in people's routines (e.g., moving, going to a government office), which we can leverage to investigate the causal effects of food environments on food outlet decisions. Our analyses focus on visits to FFOs as the key outcome because (i)

greater intake of fast food, which is typically ultra-processed, low in nutrients, and energy-dense, is a well-established risk factor for poor diets, obesity, and cardiometabolic disease[2,37]; and (ii) recent work has shown, using the same mobility dataset we utilize in this study, that visits to FFO are associated with self-reported fast food intake, obesity, and type 2 diabetes, thereby establishing the link between FFO visits observed in mobility data and nutritional health[38].

## Results

### Characterization of mobile food environments

Individuals in large urban areas travel or commute considerable distances[39], indicating that for many people, the food environments they are exposed to throughout the day are not near their homes. In our dataset, we find that the median distance from home $\mathbf{h}$ to any place visited $\mathbf{x}$ is 7.83 km (Interquartile Range, IQR, [2.47–18.63 km]), see Fig. 1. The median distance to any type of FO visited is 6.94 km (IQR [2.30 km–17.23 km]), but this varies by outlet type: the median distance to grocery stores/supermarkets is much smaller, 3.1 km (IQR [1.35 km–8.22 km]), while FFO is 6.74 km (IQR [2.50 km–16.62 km]) away (median, see Supplementary Note 1 for detailed statistical analysis of these differences). In fact, only 6.8% of the visits to FFO occur within a user's home census tract. Thus, most fast food visits occur in food environments outside of a user's home neighborhood.

To characterize a user's food environment at any given location $\mathbf{x}$, we measure the ratio of FFO to FO within a 1 km radius, $\phi(\mathbf{x})$ (See "Methods" and Supplementary Note 4 for other definitions). As shown in Fig. 1, most zones in the metro areas have small (average) values of $\phi(\mathbf{x})$. Because users move around the city, they are exposed to many different food environments: overall mobile exposure to food environments, the time-weighted ratio of FFO to FO that a user is exposed to in our 6-month entire period ($\phi_i^m$, see "Methods") has a median of 14.0% (IQR [9.7–19.0%]. We also find that users' mobile food environments are different from the food environments around their homes. Although home environments have a relatively low FFO to FO ratio (median of $\phi_i^h = \phi(\mathbf{h}_i)$ for all users is 8.2%, IQR [0–17.5%]), we find that the correlation between mobile and home environments is small $\rho(\phi_i^m, \phi_i^h) = 0.213 \pm 0.001$ across users. That correlation is slightly different across demographic groups, with residents in areas with low-income, high percentage of Black population, or large use of public transportation having a slightly stronger correlation between home and mobile environments. But in general, the correlation is small across groups $\rho \leq 0.29$ (see Supplementary Fig. S19). In other words, the food environments that users are exposed to throughout the day are different from the ones around their homes.

The ratio of FFO to FO in users' food environments is associated with various sociodemographic characteristics inferred from users' home census block group. Using linear regression models for $\phi_i^m$ (see Fig. 2, "Methods", and Supplementary Table S3 for full statistical details of these models and their comparison), we find that users exposed to mobile food environments with a higher proportion of FFO (larger average $\phi_i^m$) reside in areas with a higher proportion of residents who are Black, who have long commutes, and who have lower skill jobs, and a lower proportion of residents with more educational attainment and who depend on public transportation. Similar findings have been obtained in small surveys on the use of different modes of transport to FO and FFO outlets[19,40]. Neighborhood-level household income is not significantly associated with any mobile food environment features. These relationships differ with users' home food environments (see Fig. 2), where a greater proportion of FFO in the home neighborhood, larger $\phi_i^h$, occurs in neighborhoods with lower income, higher levels of educational attainment, and shorter commuting patterns, similar to findings done by Powell et al.[41]. Despite that, we find a stronger relationship between the socio-demographic characteristics of users and their mobile food environments ($R^2 = 0.213$ for $\phi_i^m$) than for their home food environments ($R^2 = 0.038$ for $\phi_i^h$),

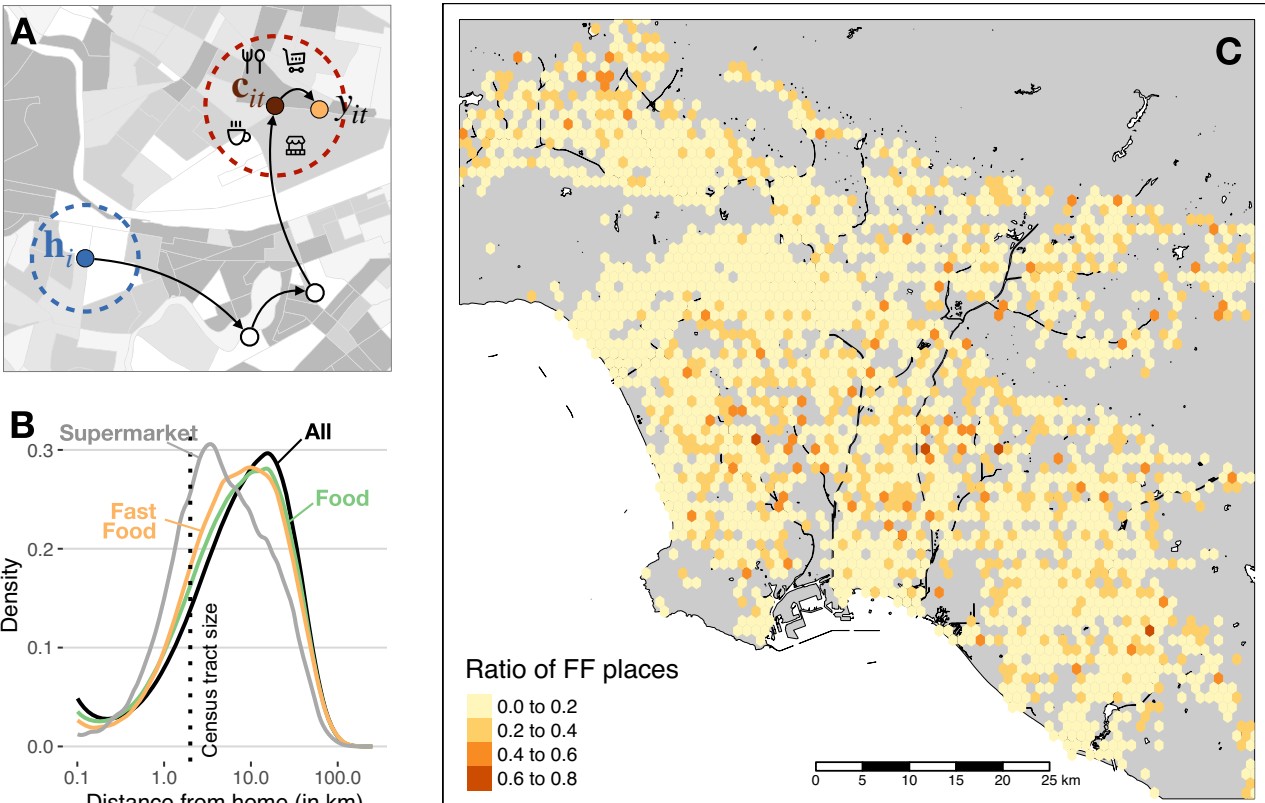

**Fig. 1 | Urban mobility conditions food environments and choices. A** in their daily life, users navigate the city from home $h$ to different places until they arrive at a context $c$ where they decide to have food in $f$. **B** Distribution of the distance from home to all visits in the city (black), all retail food outlets (green), and fast food outlets (orange). The distance traveled to food and fast food outlets is much larger than to supermarkets (gray) or the typical size of a census tract (dotted vertical line). **C** Heatmap of the ratio of fast food outlets $\phi$ in the Los Angeles metro area. The ratio is calculated within each hexagon of size ~ 1 km². Icons designed by bqlqn/flaticon.com and maps were produced in R using the publicly available TIGER shapefiles from the U.S. Census Bureau[69].

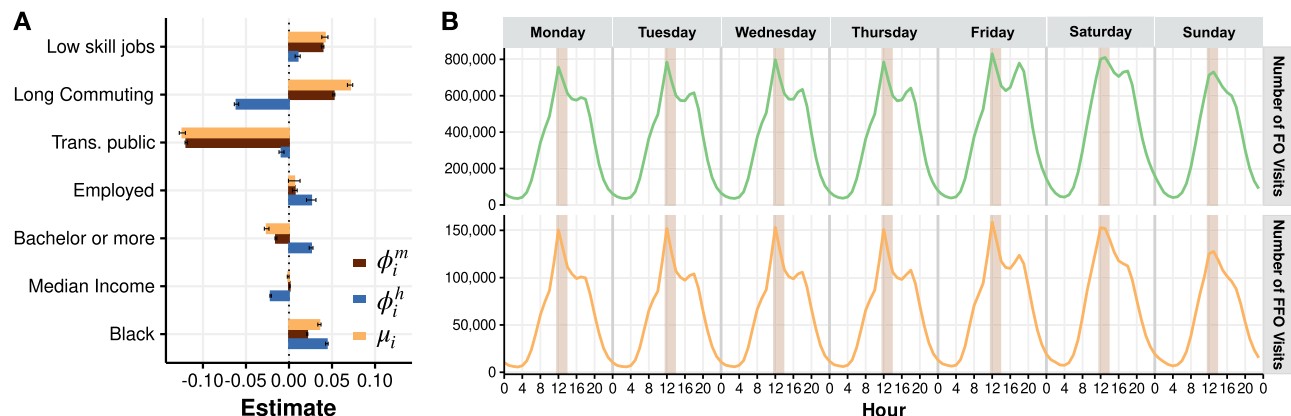

**Fig. 2 | Demographic and daily patterns of fast-food environments and visits. A** Relationship between FF environments and socio-demographic traits. Bar shows the coefficient estimates for OLS regression models of overall mobile and home FFO environments ($\phi_i^{m,h}$) and the fraction of visits ($\mu_i$) by user for the proportion of workers in low-skill jobs, the proportion of people with long (>45 min.) commute, the proportion of people taking public transportation for commuting, the proportion of people employed, the proportion of people with higher education level, median household income and proportion of Black people in their home census block group. Error bars are 95% confidence intervals for those coefficients. See Supplementary Table S3 for more details. **B** Daily patterns of the number of FO (top) and FFO (bottom) visits in our urban areas. The shaded area corresponds to the lunch observation period taken to determine the action $y_{it}$ in our model (1).

see Supplementary Table S3 for more details. Similar results are found for more complex non-linear models (see Supplementary Note 7). This suggests that socio-demographic differences propagate slightly more strongly to people's experienced mobile food environments than to their food spatial accessibility at home, possibly because those demographic traits slightly shape the differential access to different environments, targeted marketing, and other social and structural forces.

## Who, when, and how much people visit FFO
While $\phi_i^{h,m}$ describes the home and mobile exposure to fast food options at any given moment, we encode the actions of users at time $t$

with the variable $y_{it}$. If individual $i$ chooses to visit an FFO among the FO options, then we set $y_{it} = 1$. If they select a non-fast food option, then we set $y = 0$. The overall averaged fraction of FFO choices to FO options $\mu_i = \overline{y_{it}}$ (see "Methods") in all environments over our observation period is heterogeneous across users with a median of 0.133, IQR [0.025, 0.273] (See Supplementary Note 1). That is, 13.3% of visits to FO are to an FFO (median). Our results also show that a significant proportion of users never visited FFOs (22.9%) during the 6-month period of observation. Using similar regression models as before, we find some statistically significant differences in the ratio of visits to FFO ($\mu_i$) across demographic groups. Similar to the results described above, individuals visit FFOs more often if they live in areas with less use of public transportation, with a higher proportion of Black residents, with longer commutes, and with less educational attainment. Similarly, income has a smaller association with the ratio of FFO visits when compared to the other demographic traits. Our results align well with the contradicting evidence of little variability in fast food intake across income levels despite consistent differences based on educational levels and race and ethnicity[42–44]. Additionally, we find that traditional demographic traits–race and ethnicity, type of job, income, and educational level–have a weaker association with fast food visits than characteristics related to mobility and time constraints–the use of public transportation and long commuting (see Supplementary Table S3 for full statistical details of these models and their comparison). However, it is important to note that the explanatory power of this association between $\mu_i$ and socio-demographic variables is low, $R^2 = 0.052$. Thus, even though we find statistically significant differences, our results suggest that overall FFO visits do not meaningfully differ across different socio-demographic groups. Many types of people visit FFO in urban areas.

We also found that most food outings happen between midday (lunchtime) and the evening (dinner time), both during weekdays and weekends, see Fig. 2. FFO visits have the same temporal pattern, with a peak of visits to fast food happening around lunchtime from Monday to Sunday.

### Relationship between mobile food environments and fast food visits

To understand the effect of food environments on fast food visits, we first study the relationship between total average exposure to fast food $\phi_i^{h,m}$ and the overall observed ratio of FFO to FO visits, $\mu_i$. Many studies with small datasets have found null or contradicting results regarding the association between total exposure to FFO and fast food intake[32,45]. We find a positive relationship between a user's average daily exposure to FFOs within their mobile food environment, $\phi_i^m$, and overall observed ratio of visits of FO to FFO, $\mu_i$. Specifically, the correlation between these two variables is $\rho(\phi_i^m, \mu_i) = 0.268 \pm 0.001$. However, the correlation between FFOs within a user's home food environment and overall ratio of visits to FFO is weaker, with $\rho(\phi_i^h, \mu_i) = 0.068 \pm 0.001$. These results are expected, given that most food outings happen far away from home, and suggest that an individual's exposure to FFO across the day, rather than within their home environment, is a more important driver of the decision to get fast food.

To better understand what drives the association between FFO exposure and visits, we need to go beyond total exposure. Visits to FFO might be due to individual preferences, to structural or social cues received from frequent long-term exposure to environments high in FFO[14], but also might be a direct response to the food environment where those decisions are made. A person may choose to visit different food outlets in food environments with different features. At the same time, different people exposed to the same food environment may make different decisions about which food outlet to visit. To identify the association between food environment features and FO decisions, we first design an individual analysis of each visit to a FFO. We restrict the data to FO visits during lunch hours (from 11 h30 to 14 h local time)

because this time window has the highest FO and FFO visits (see Fig. 2). Most importantly, time constraints at this time of day could make the food environment options experienced before going to get lunch even more relevant. Indeed, we find that the relationship between action $y_{it}$ and the food environment features is stronger around noon than at other times of the day (see Supplementary Note 7).

For decisions about food outlet visits $y_{it}$ at time $t$, we define the context of that decision as the last place in the morning (until 11:30) where a user was observed (context $\mathbf{c}_{it}$), and characterize the food environment in that context by $\phi(\mathbf{c}_{it})$. Thus, the context for the food outing is the environment where the user was before the outing. To ensure that the context offered a choice between FFO and non-FFO, we focus exclusively on observations where the context $\mathbf{c}_{it}$ contains both types of outlets. We have extensively checked that our results do not depend on the precise definition of the context and of the food environment around it (see "Methods" and Supplementary Note 12). We use a logistic regression model to estimate the impact of user $i$'s context environment on their decision to visit an FFO vs. a non-FFO at time $t$:

$$\Pr(y_{it} = 1) = \text{logit}^{-1}\left[\beta_0 + \alpha_i + \delta_t + \beta\phi(\mathbf{c}_{it})\right] \qquad (1)$$

where $\text{logit}^{-1}(x) = e^x/(1 + e^x)$, $\alpha_i$ is a fixed effect accounting for individual average preferences to visit FFO, $\delta_t$ is a fixed effect accounting for daily variation, and $\phi(\mathbf{c}_t)$ is the ratio of FFO in the food environment a user was exposed to prior to the visit. By imposing a fixed factor by user, we separate the effect of individual preferences on visits to FFO from the effects of the food environment. For our regression, we only consider users that went to both fast and non-fast food outlets at lunch at least once during the 6 month observation period and where the food environment they were exposed to before lunch included both FFO and non-FFOs.

Our results, illustrated in Fig. 3, show a large association between features of the mobile food environment and visiting an FFO. The model produces a log-odds of $\beta = 1.87 \pm 0.033$ for all FO visits at lunchtime: when the context includes 10% more FFO, there is an increase in the odds to visit an FFO of $(e^{\beta \times 0.1} - 1) \times 100 \simeq 20\%$. This influence of the food environment one is exposed to before going to lunch was similar during weekdays and weekends and at different times of the day (not only during lunch hours) (see Fig. 3 and Supplementary Note 7). Additionally, the magnitude of the effect is largely independent of an individual's income (see Fig. 3), other socio-demographic traits, distance from home (see Supplementary Note 7) or metro area (see Supplementary Note 10). Finally, as expected, we find that distance between context and lunch choice modulates this effect, although, for the majority of our data, we still get that the impact of context is positive and significant. We also see similar results for other food choices like Asian or Latin-American food (see Supplementary Note 11). Thus, individuals with different socio-demographic backgrounds appear to respond similarly to mobile food environments at different times.

Despite finding an effect of mobile food environments on visits to FFO, it could be that the lack of non-fast food options predominantly affects individuals when they are in a new place. This may be because they are less equipped to navigate the new food environment and identify food options they prefer. It is also plausible that a lack of options for different FO types constantly affects users who would otherwise visit FFO with a lower frequency, given their individual preferences. To address these questions, we propose a semi-causal framework using a natural experiment to investigate the relationship between habitual FFO context and FO decisions. In this experiment, we observe people who changed their quotidian context during the study. Using the time series of the different contexts before lunch and changepoint analysis, we were able to detect a small fraction of users (0.46%) that changed their habitual context before lunch within our

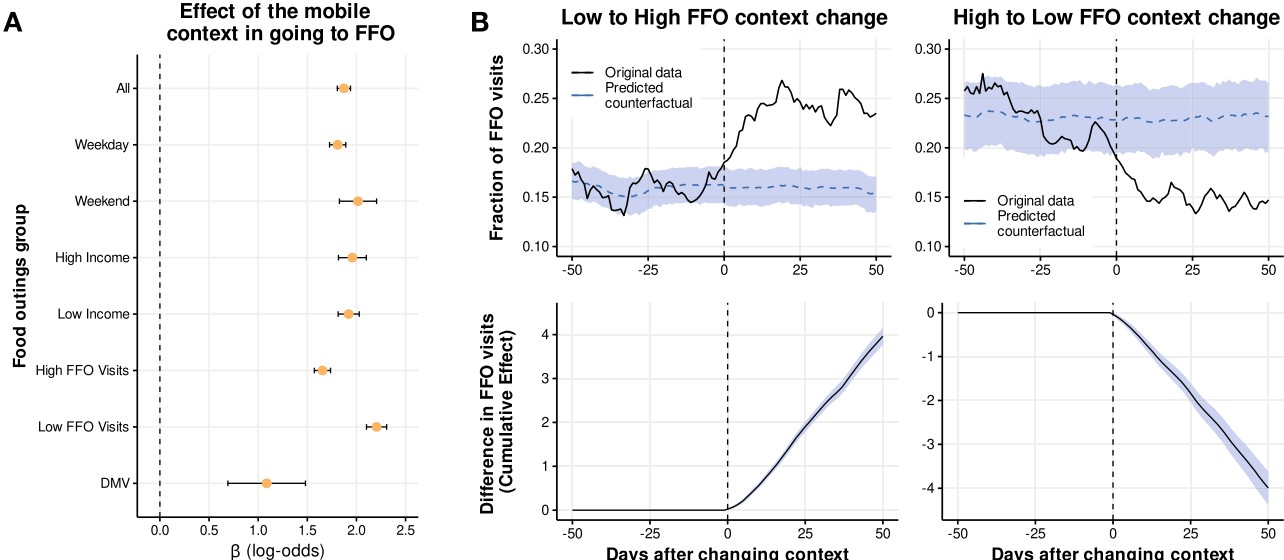

**Fig. 3 | Effect of mobile food environments on fast-food visits. A** Effect of the mobile food environment on visiting a FFO at different times, locations, or for different income or FFO visitation groups. Values show the coefficient $\beta$ of the logistic regression in Eq. (1) for the food visits (outings) corresponding to the different groups. Error bars are 95% confidence intervals for those coefficients. See Supplementary Table 4 for more details. **B** Evolution of the fraction of FFO visits (top) and cumulative difference in FFO visits (bottom) for groups of users that change their contexts from Low to High FFO environments (left) and High to Low FFO environments (right). The dashed horizontal line is the predicted counterfactual for groups of users that changed their context. The shaded area corresponds to a 95% confidence interval for that predicted counterfactual.

observation period (see "Methods" and Supplementary Note 5 for details about this detection). Due to our large sample size, this small fraction had a large enough size (approx 8.5k users) to do statistical inference. Those users were split into four groups depending on whether they changed to a context with similar or different low ($\phi < 0.13$) or high ($\phi > 0.13$) exposure to fast food. We found that around 35% of the users who changed their context remained exposed to very similar food environment features before and after (High→High FFO exposure or Low→Low FFO exposure). Around 15% of the users changed their context, such that the FFO in their typical pre-lunch food environment went from Low→High exposure to FFO, and another 15% from High→Low FFO exposure. We construct the time series of the fraction of times that users visit an FFO relative to the day when they change contexts. We study the impact of the change on FFO visits using Bayesian structural time-series models (see "Methods") by comparing the group that changed their FFO contexts (Low→High and High→Low) with the counterfactual of those that, despite changing their context, were exposed to similar FFO food environments (Low→Low and High→High, respectively, see "Methods"). Results are presented in Fig. 3, which shows that the group that changed from Low→High FFO exposure increased their fraction of FFO visits from ~16% to ~25%. Similarly, users that changed their context from High to Low FFO exposure decreased from ~24% of FFO visits to ~15%. The counterfactual of users that changed contexts but remained exposed to food environments with similar FFO ratios maintained a similar fraction of FFO visits. We also note that this effect is statistically robust and persistent, remaining even 50 days after changing their context. In cumulative numbers, we find that users who changed their context to High (Low) FFO exposure visited FFO 4 times more (less) in 50 days than those who remained in food environments with similar FFO ratios. These results suggest that the effect of the food environment is strong even for the same users subject to different habitual contexts, and it is not only driven by visits to new places.

Finally, we sought to analyze a setting as close as possible to one where people are placed in random locations within the city. Studying food decisions under such circumstances would preclude potential estimation bias from omitted variables affecting both food

preference and location in the city. For this, we propose a natural experiment looking at people who visited the Department of Motor Vehicles (DMV), the food environment they faced there, and the decision they took to get a meal. In particular, we repeat the analysis made for Eq. (1) but only when the context before lunch is the DMV. These locations are commonly visited for obtaining a driver's license, government ID, voting, and other services. Moreover, the food environment around a DMV is unlikely to be a determining factor when choosing a DMV location, compared to other factors such as availability of appointments and distance from home/workplace. While distance to home/workplace can be influenced by latent factors such as income and even food preferences, the set of DMVs in our cities is small enough that many people need to exit their home and work neighborhoods to go there. Moreover, the time constraints caused by the scarcity of appointments help make the choice of DMV location less determined by spatial accessibility alone. Another important factor to consider was whether people going to the DMV will indeed look for lunch around it. It is possible that some of the customers of the DMV decide to eat before heading to it, or after returning to their work or home. We solved this problem by computing the probability of going to fast food given that the person went to a food outlet. This filtered out people who belong to the aforementioned group.

In our dataset, we detected 53k visits to the DMV across a 6-month period. The median distance traveled from home to the DMV was 7.62 km (IQR, [3.48–15.44 km]), greater than the median distance to FO. We consider the DMV as the context of a FO visit if such a visit occurs within 2 hours of the DMV visit. To investigate the effect of the DMV food environment on $y_{it}$, we use a logistic regression model $\Pr(y_{it} = 1) = \text{logit}^{-1}[\beta_0 + \hat{\alpha}_i + \phi(\mathbf{c}_{it})]$ similar to Eq. (1) model. Since we typically have one visit to DMV per user, we model individual preference $\hat{\alpha}_i = \mu_i$ as the fraction of visits to FFO of user $i$, and we do not include daily fixed factors. The effect of the DMV food environment is shown in Fig. 3. We find a significant effect, although a little smaller in size than the effect of exposure to habitual contexts, with log-odds of $\beta = 1.09 \pm 0.20$. This third analysis corroborates that features of food environments influence FFO visits.

## Policy implications

The observed relationship between food environments with high ratios of FFOs and increased visits to FFO, specifically for mobile food environments, implies that more targeted interventions to reduce visits to FFO can be designed. Many intervention approaches have focused on improving food environment quality around the home neighborhood or in geographic regions with poor food environments, without accounting for where people more frequently visit food outlets. Notable examples include the over one billion dollars leveraged by the U.S. Healthy Food Financing Initiative to finance healthy food retail in under-served local neighborhoods[21], and the 'fast food ban' implemented in 2008 in neighborhoods in South Los Angeles with a high prevalence of FFO[22]. Our findings highlight that FFO visits often take place well beyond the home neighborhood, and suggest that strategies that solely focus on geography and spatial access to food outlets in the home neighborhood (ignoring human behavior) are likely to lead to sub-optimal intervention effects. Indeed, evaluations of major policies and interventions to improve the quality of neighborhood food environments have demonstrated they have little impact on diet or diet-related diseases[25,46]. Here, we use the results of our observational study to identify the optimal locations to intervene in food environments to have the greatest impact on decreasing FFO visits. Specifically, these will be contexts demonstrating the highest ratios of FFO to FO, the highest frequencies of user exposure and FFO visits, and the largest observed impact of food environment features on a population's FFO decisions. We investigate the likely effects of intervention strategies that change the ratio of FFO to FO in these optimal impact locations vs. interventions targeting locations such as neighborhood food deserts and food swamps, the traditional choice locations for intervention.

Changing the ratio of FFO to FO in an area can be accomplished by various strategies or interventions, including: decreasing the number of FFO relative to non-FFO, increasing the number of non-FFO relative to FFO, or converting a FFO into a non-FFO. Policy strategies in practice are more likely to involve one of the first two approaches, such as implemented interventions that have banned new FFO from opening in a specific area[22,47,48], or supported non-FFO to obtain business licenses or opportunities in food retail[49]. The former has been implemented in Los Angeles[22] and in northeast England, where a blanket ban in a district led to a reduction of 15% in $\phi(x)$[47]. The latter strategy was recently exemplified by a US White House nutrition initiative that invested in programs supporting local entrepreneurs to open healthy prepared-food outlets[50]. The last approach, converting a FFO to a non-FFO, could be seen as similar to approaches to shift the balance of the healthfulness of food items being sold at existing FFO[51].

Assuming that our intervention I changes the context in an area $\Omega$ by $\delta\phi/\delta I$ and that users are still making their decisions according to the model in Eq. (1), the change in the number of FFO visits made immediately after being exposed to the food environment of $\Omega$ can be obtained as (see "Methods"):

$$\Delta^{\mathrm{FFO}}(\Omega) \simeq \sum_{\mathbf{c}_{it} \in \Omega} \beta \frac{e^{X_{it}}}{(1 + e^{X_{it}})^2} \frac{\delta\phi}{\delta I}, \qquad (2)$$

where $X_{it} = \beta_0 + \delta_t + \alpha_i + \beta\phi(\mathbf{c}_{it})$. This expression shows that the effect of an intervention in an area $\Omega$ depends on three factors: (i) the susceptibility of FFO visits with respect to the availability of FO options in the area $\Omega$, expressed through $e^{X_{it}}/(1 + e^{X_{it}})^2$ and ultimately by the balance between individual preferences $\alpha_i$ and the context $\phi(\mathbf{c}_{it})$, (ii) the number of decisions made in area $\Omega$, expressed by the sum, and (iii) the effect of the intervention on the context $\phi$, expressed by $\delta\phi/\delta I$. For example, for the same intervention $\delta\phi/\delta I$, we can have an area where many people go, but they have a large individual preference for FFO ($\alpha_i \gg 0$), causing the effect of the intervention to be small because they are not influenced by the food

environment features. On the contrary, we can have an area that many people do not visit, but the people who do visit are highly influenced by that context's food environment and the number of FFO around ($\alpha_i \simeq 0$). An intervention in this latter area can meaningfully change a number of food outlet decisions.

To illustrate this, we consider a simple intervention in which we decrease the ratio of FFO to FO by one unit in a particular area. In this case, $\phi(\mathbf{c}_{it})$ changes by approximately $\delta\phi/\delta I \simeq -1/n_\Omega$ where $n_\Omega$ is the number of FO in the area. We see this strategy as a theoretical illustration of the general concept of improving the food environment in an area by shifting the balance of FFO to healthier outlets using any of the approaches discussed above, by the same amount of effort per area. If the methodology we develop in the following were to be used, this could be implemented by setting the derivative of $\phi$ in our formula to be the expected derivative of the intervention. We have also extended the model to all times of the day and week to describe the full effect of the strategy (see "Methods"). Assuming that we have limited resources to change 100 food outlets, where are the areas in which our intervention maximizes its impact? Here, we compare four different strategies.

In the first strategy (Food Swamp intervention), we select the areas with the largest (average) values of $\phi$, i.e., the areas where FFO predominate. For comparison and to resemble prior food desert interventions around home neighborhoods, in our second strategy (Low Food Access intervention), we select the areas that have the largest values of $\phi$ and are classified by the USDA as food deserts (both low-income and low-supermarket-access)[8]. The third strategy (Food Hotspots intervention) is implemented by selecting the areas where most FO visit decisions are made. However, these strategies do not incorporate individual preferences or susceptibility to food environments. Thus, in our fourth strategy (Behavior-Environment intervention), we select areas $\Omega$ as the top areas ranked by $\Delta^{\mathrm{FFO}}(\Omega)$ in Eq. (2), which includes not only the context but also the individual preferences of people deciding in those contexts. Figure 4A shows the relationship between the change in FFO visits $\Delta^{\mathrm{FFO}}(\Omega)$ and the average context $\phi$ in the different areas (census tracts) in our cities. The figure illustrates the dependence between these two variables, but there is still significant variability. For the same average of $\phi$, we have areas where the change in $\Delta^{\mathrm{FFO}}(\Omega)$ spans two orders of magnitude. In this representation, our Food Swamp and Behavior-Environment interventions are very easy to interpret. They consist of choosing the rightmost (greatest change in context) or topmost (greatest change in FFO visits) areas, respectively. In the case of Food Swamps and Low Food Access interventions, we can see those strategies choose areas $\Omega$ in which $\Delta^{\mathrm{FFO}}(\Omega)$ is small because not many decisions are made (Low Food Access) or because users are less affected by the FFO environment in those areas (Food Swamps). The Food Hotspots intervention chooses areas where the most FO decisions are made, but without considering whether users are affected by the FFO environment. As a result, the total effect of the four strategies is very different; see Fig. 4B. Overall, the Behavior-Environment intervention would be 1.93× to 3.85× times more efficient in decreasing FFO visits than interventions that used only the FFO context where decisions are made or around the home neighborhood. In relative terms, by changing one hundred FFO (0.22% of the total), our Behavior-Environment intervention could avert around 0.56% of the visits to FFO, while other strategies could only affect 0.29% of those visits at most. If we scale these numbers to the total population, our Behavior-Environment could avert around 719k visits to FFO in 6 months, compared with only 442k at most in the other interventions. Furthermore, the impact of the interventions is predominantly independent of income (see Fig. 4B), health conditions (see Supplementary Note 9), or particular city (see Supplementary Note 10), and thus its effect does not concentrate on particular groups or geographies. For example, we found that our Behavior-Environment intervention is still 2× to 2.5× more effective than the rest of the interventions at targeting

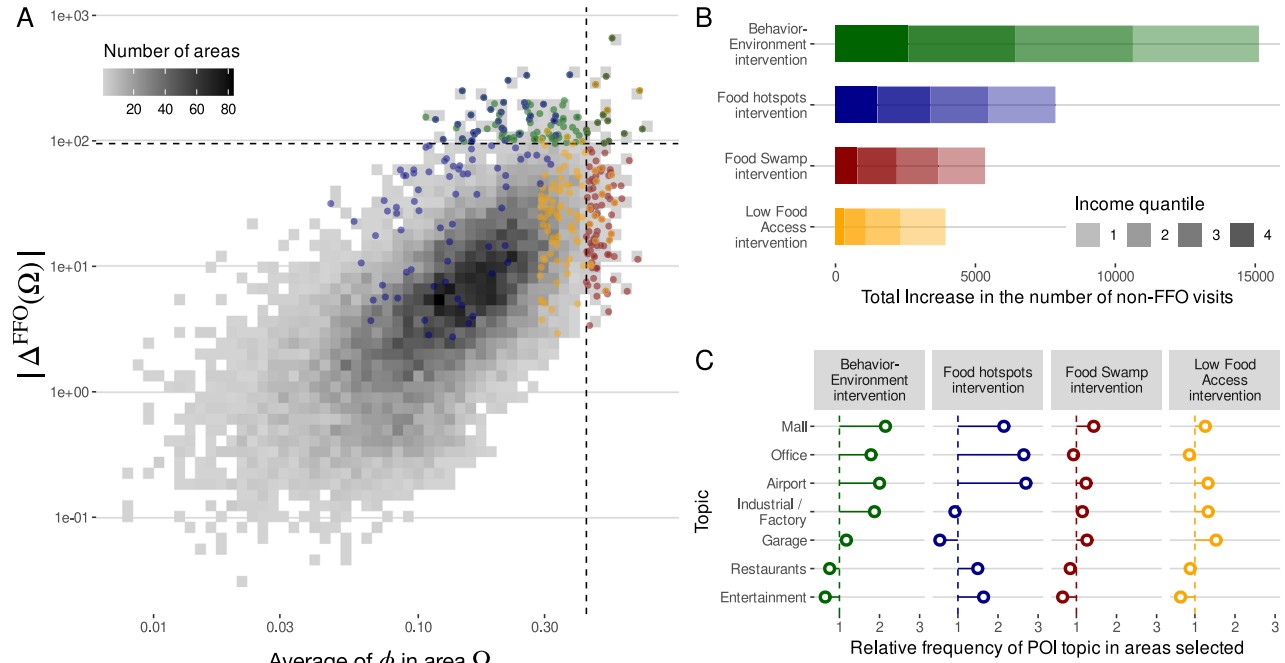

**Fig. 4 | Impact of different interventions on food environments in visits to fast food places. A** Change in the number of visits to FFO in an area $\Omega$ after deploying an intervention I as a function of the ratio of FFO to FO, $\phi$, in that area. Dashed lines marked the thresholds for the strategies to choose the top 100 areas by FFO ratio (vertical line, red points, Food Swamp intervention) or by the total change in the number of visits (horizontal line, green points, Behavior-Environment intervention). Orange points are those 100 areas chosen as the top home areas according to the FFO environment (Low Food Access intervention), and blue points correspond to those 100 areas chosen as the top areas containing more FO visits (Food Hotspots intervention). **B** Total effect of each intervention strategy in the different urban areas. Shades correspond to the number of actions changed by different income quantiles. **C** Relative frequency (to all areas) to find different groups of POIs (topics) in the areas selected for each intervention strategy.

decisions made by high obesity or diabetes prevalence groups (see Supplementary Note 9).

Finally, to understand what types of areas $\Omega$ are targeted in our Behavior-Environment intervention, we use latent topic analysis to determine the groups of points of interest (POIs) that appear more frequently in those areas (see Supplementary Note 8). As shown in Fig. 4, we find that the groups of POIs related to Malls, Industry/Factory, Airport or Office are more likely to appear in our targeted areas than in the rest of the areas in the city and the rest of interventions. Other groups of POIs, like Restaurants or Entertainment, are less likely to be areas selected in our Behavior-Environment intervention but more likely to be selected in the Food-Hotspots intervention. However, this intervention is less efficient, meaning that despite a lot of food visits happening around Restaurant and Entertainment areas, people making food decisions there are less affected by the environment, and thus the intervention is less successful. In summary, these large differences in our results suggest that more efficient interventions could be deployed to specific types of food environments away from residential areas, where FO decisions are most influenced (and likely constrained) by the environment, i.e., closer to work, travel, or shopping areas. Behavioral interventions in cities with different relative compositions of those areas might have slightly separated effects. For example, although behavioral interventions are always more effective than static interventions, the difference is slightly smaller in cities like Philadelphia, Seattle, or Los Angeles, suggesting that behavioral interventions depend somewhat on city-specific structural or commercial characteristics (see Supplementary Note 10).

## Discussion
The effect of food environment exposure on diet and related diseases has been studied extensively; however, in many cases, small, limited datasets have hampered the ability to understand this complex process in urban areas[18–20,22,23,25,27,30–33,45]. A major research limitation has been a focus on cross-sectional observations of static food environments around the home, and limited study of the mobile food environments people are exposed to and food outlets they visit as they navigate their day-to-day environments[20,30–32,45]. Our dataset and semi-causal study designs have allowed us to analyze the effect of exposure to food environments on food choice at a remarkable granularity and across diverse populations. This longitudinal, individual-level behavior data has also allowed us to analyze how food choice is motivated by features of food environments people are exposed to in their daily routines vs. by individual preference. We find that most visits to fast food outlets occur relatively far (a median of 6.74 km) from home and that exposure to low-quality food environments beyond the home is significantly linked to increased visits to FFO, across diverse socio-demographic groups. Furthermore, our results show that the composition of mobile food environments also affects the choice to visit other types of food options than fast food (e.g., Asian and Latin American cuisines; see Supplementary Note 11), reinforcing the role of mobile food environments beyond the home on multiple types of food decisions.

Previous policy interventions to food environments have been targeted at low-quality neighborhood food environments, such as 'food swamps' characterized by an abundance of fast food options. Interventions focused on static features of neighborhood environments do not reflect the complex intertwined process of human mobility, food environment exposure, and eating decisions in urban areas. This mischaracterization may partially explain the unsuccessful neighborhood interventions that aim to regulate the fast food environments near where people live[22,23,34]. To reduce visits to FFO and improve overall dietary quality, our results suggest that we may need to intervene in the mobile food environments that are not only characterized as low-quality, but also where most FFO decisions happen,

and, importantly, where people have been demonstrated to be most influenced or constrained by the options available in that area - observations possible with this mobility data. The most efficient interventions may be further from people's homes, in areas where food environments are more determinant to food decisions, like work, school, travel, or shopping areas. Finally, future policy-specific work should seek to investigate further details of the operationalization and implementation of the theoretical strategies we investigate here for shifting the balance of FFO to non-FFO in a food environment to optimize the dietary health of individuals frequenting that environment, including specifically how and what kinds of outlets should be involved in the interventions. Mobility data is increasingly available for researchers and policymakers, and could be used as part of 'smart city' initiatives to identify a community's food hot-spots and tailor their food environment interventions by including behavior analysis of their actions when they move around the city.

While food environment interventions based on these factors were demonstrated to impact all groups equally, including lower income groups and those with higher rates of diet-related disease, these data also allow us to observe vast inequities in exposure to food environments of lower nutritional quality for historically marginalized communities. Based on these observations and innumerable studies demonstrating the structural inequities between socioeconomic groups food environment exposure and access[9,11,12,14], future targeted interventions should be designed to account for the additional and complex dimensions of fairness and equity[52,53], while accounting for individual preferences and projected decisions.

In addition to identifying locations for food environment intervention, our methodology could also be used to inform individual-level interventions promoting or encouraging visits to food outlets located in food environments that have more diverse, healthy options; for example, through a mobile app. Our results show that the impact of mobile food environments is also quite strong for other types of foods than fast food. Thus, the design of holistic individual-level interventions would require a combination with other data about food intake (e.g., delivery, nutrients[54], degree of food processing[55]), healthiness of available menu options[56], food preference and sentiment[57,58], and price sensitivity[46,59,60].

Population-scale mobility data provide useful, dynamic behavioral indicators of FF visits and consumption[35,38], re-defining static notions of food deserts or food swamps to mobile food environments determined not only by the diversity of FO available in those environments but also by their frequency of use and people's susceptibility to what they offer. We hope our results and our complex-systems methodology using large-scale mobility data can inform more efficient policies and interventions on food environments complementing and extending those around home neighborhoods[54,61–63] or efforts to increase the healthfulness of food items being sold at FFO[51].

Our study has several limitations. Although it is well established that eating at FFOs is linked to poorer diet quality[28], and there is a strong association between observed visits to FFO and FF intake[38], foods of diverse nutritional quality are sold across FFO[56]. We have currently not examined the extent to which healthier options are offered at the FFO visited by our sample, nor how these options may impact purchase and consumption behavior. On the other hand, although it is likely that visits to FO that last more than 5 minutes lead to an eventual purchase, which is furthermore supported by the strong association between observed visits to a particular type of food outlet (fast food) and intake of food of that type[38], there is no complete guarantee. Our results, therefore, serve as a proxy and are a lower bound for individuals' potential FF intake. Also, since visits are attributed to the closest POI, there are limitations to the detection of visits to certain food outlets, such as those in multi-story or multi-purpose buildings (e.g., malls) where FFO are frequently found. Additionally, because we only detect visits greater than five minutes in duration, we may miss very brief FF outlet visits (e.g., drive-thrus). Finally, although our casual framework provides robust evidence about the impact of mobile food environments on people's FFO visits, we believe our results may be further tested through carefully designed experiments and interventions. These experiments and interventions should also explore potentially different effects of food environments on other types of food outlet visits (e.g., visits to full-service restaurants or grocery stores). Finally, our mobility data sample from 2017 may not reflect changes in exposure to and impact of food environment on FO visit behavior following the pandemic and related changes that have occurred in the intervening years, including increased time spent in home neighborhoods and the great expansion of food delivery apps and their coverage. However, in settings where individuals are constrained by their environment, our findings and population-scale mobility framework likely still apply.

## Methods
### Data
We use individual-level anonymized mobility data of 1.86 million anonymized users in 11 US metropolitan areas over a period of 6 months, from October 2016 to March 2017. The mobility data were collected with the informed consent of the users, who opted-in to anonymized data sharing for research purposes under a GDPR and CCPA-compliant framework. Our mobility data were obtained from Spectus, a location intelligence and measurement company. Since the data used was anonymized and spatially aggregated at places, categories, or census areas, we were granted an Exemption by the MIT Committee on the Use of Humans as Experimental Subjects (COUHES protocol #1812635935) and its extension #E-2962. To identify visits to FO and FFO, we extracted from the mobility data the stays (stops) of people around a large collection of points of interest (POI) obtained from Foursquare, see Supplementary Note 1. FFOs are quick-service restaurants where patrons typically pay before eating and were defined using Foursquare's taxonomy and a name search using a list of chain FFO; FOs, which represent all retail food outlets including grocery stores, supermarkets, big box stores, convenience stores, restaurants, were based on Foursquare's existing taxonomy (see Supplementary Note 3). We have comprehensibly checked that our results do not depend on the choices made on the definition of stays, the categorization of the FFO, the POI database, or the definition of the environment and the population representativity of our data. A full description of those definitions and robustness checks is provided in the Supplementary Notes 1, 2, 3, 4, and 12. Census demographic data come from the American Community Survey (5y) from the Census[64]. Census tract-level estimates for different health risk behaviors and outcomes were obtained from the 2017 edition of the PLACES Local data for Better Health dataset from the CDC[65]. Food desert data come from the Low-income and low-supermarket-access classification of census tracts done by the U.S. Department of Agriculture in their Food Environment Atlas[8]. Further details about those datasets can be found in the Supplementary Note 1.

### Definition of the home and mobile food environment and food context
To characterize the food environment users are exposed around a given place $\mathbf{x}$ we measure the ratio of FFO to any FO within a 1km radius, $\phi(\mathbf{x})$. We have extensively checked that our results do not depend on other definitions of the food environment. For example, in Supplementary Note 12 we show that similar results are obtained when we take $\phi(\mathbf{x})$ as the ratio of FFO to FO of the closest 25 FO to $\mathbf{x}$, a definition that accounts for the different density of FO around the city. Home food environments are described by the value of $\phi$ around where people live $\phi_i^h = \phi(\mathbf{h}_i)$. While food environment exposure around place $\mathbf{x}$ is given by $\phi(\mathbf{x})$ we also computed the total exposure a user gets by moving around as $\phi_i^m = \sum_t \tau(\mathbf{x}_{i,t})\phi(\mathbf{x}_{i,t})/\sum_t \tau(\mathbf{x}_{i,t})$ for all times $t$ the

user stops for more than five minutes (irrespectively of whether a FO is visited) and where $\tau(\mathbf{x}_{i,t})$ is the duration of the stop of individual $i$ at $\mathbf{x}_{i,t}$. The food context is the environment where the user was before going to visit a FFO. If the user was at $\mathbf{c}_{it}$ before lunch, the context is measured by $\phi(\mathbf{c}_{it})$. Finally, the overall averaged fraction of visits to fast food of individual $i$ is computed as $\mu_i = \overline{y_{it}} = \sum_t y_{it}/N_i$ for all times $t$ the user visits a FO and where $N_i$ is the total number of FO visits of individual $i$.

### Statistical models

To test the effect of mobile food environments, we have run a number of statistical models. For the main results in Fig. 2, we used logistic regression to link the binary output $y_{it}$ to the ratio of FFO options around the context $\phi(\mathbf{c}_{it})$, see Eq. (1). We control individual preferences and daily patterns by introducing a fixed effect by user ($\alpha_i$) and day ($\delta_t$). Regression was only performed for those individuals that have at least one FFO and non-FFO visit. To account for potential heterogeneity in our regression, we also cluster errors by day and user. Similarly, for the visits to the DMV, we used a simpler logistic regression. Since we typically have only one observation and day per user, the fixed factor $\alpha_i$ was substituted by the actual observed fraction of visits to FFO of each individual, and we dropped the daily fixed factor. Finally, for the analysis of the different interventions, we have extended the model (1) to the rest of the day by considering each stay within our dataset as a context $\mathbf{c}_{it}$, and we evaluate if there is a food visit $y_{it}$ in the next two hours after that stay. A full description of those models, their predicting power (accuracy around 80% at individual level and 90% at area level), and the impact of distance between context and action are provided in the Supplementary Note 7.

### Detecting and analyzing the change of context

To identify those users that change their context before lunch, we have used a statistical methodology to detect change points in time series (see Supplementary Note 5). Using this method, we detected 7913 users in our dataset that changed context during our observation period. To provide a statistically robust estimation of the impact of that change in FFO visits at lunch, we define four groups of users depending on whether their contexts before/after the change have Low ($\phi < 0.13$) or High ($\phi > 0.13$) ratio of FFO in their contexts. We investigate the FFO ratio of visits of those groups of users that change from Low to High and from High to Low using those that change from Low to Low and from High to High as counterfactuals, respectively. Note that we did not use as counterfactual those users that stayed in the same geographical context, but only those that changed their geographical context. This was done to reduce the possibility of some endogeneity between changing contexts and the food environment in the previous context. To analyze the difference in response to the change, we use Bayesian Structural Time Series to predict how the response would have evolved after the change to a different context if the change had never happened[66]. Further details about this methodology can be found in the Supplementary Note 5.

### Interventions

To investigate the effect of an intervention strategy in an area $\Omega$ we evaluate the change in the probability $P(y_{it}=1)$ for each action with context $\mathbf{c}_{it}$ in that area using the extension of the model in Eq. (1) for the full day with and without the intervention I. The total increase in the number of non-FFO visits can be approximated by the derivative of the model in Eq. (1):

$$\Delta^{\text{FFO}}(\Omega) \simeq \sum_{\mathbf{c}_{it} \in \Omega} \frac{\delta \Pr(y_{it}=1)}{\delta I} = \sum_{\mathbf{c}_{it} \in \Omega} \beta \frac{e^{X_{it}}}{(1+e^{X_{it}})^2} \frac{\delta \phi}{\delta I} \qquad (3)$$

where we are assuming that we only change the food environment $\phi$ in the intervention I. This expression is evaluated for a fixed amount (100) of areas $\Omega$ chosen by different criteria in each intervention strategy. See

Supplementary Note 13 and 9 for further details about how interventions are defined and how we evaluate their impact on FFO visits.

### Topic analysis of the areas

To identify the type of areas where the most efficient interventions happen, we use topic modeling to describe the different groups (topics) of POIs in each category that co-occur in the 18896 census tracts in our urban areas. Using Latent Dirichlet allocation (LDA), we found 20 groups of POIs and analyzed their composition. The topics are easily recognizable (see Supplementary Fig. S11), and we manually annotated them as Airports, Malls, Office, etc. Each census tract can be described then by the frequency of each of the 20 topics within it. Further details about this methodology can be found in the Supplementary Note 8.

### Reporting summary

Further information on research design is available in the Nature Portfolio Reporting Summary linked to this article.

## Data availability

The data that support the findings of this study are available from Spectus through their Social Impact program https://spectus.ai/social-impact/, but restrictions apply to the availability of these data, which were used under the license for the current study and are therefore not publicly available. Anonymized data to reproduce the results of our paper is available on request from https://doi.org/10.5281/zenodo.7798632[67]. Other data used come from the American Community Survey (5y) from the Census (https://www.census.gov/programs-surveys/acs), the PLACES Local data for Better Health from the CDC, or the Food Environment Atlas from the U.S. Department of Agriculture (https://www.ers.usda.gov/data-products/food-environment-atlas) which are publicly available on their websites. A description of these datasets is given in Supplementary Note 1. Source data are provided with this paper.

## Code availability

Code to run the analysis has been deposited on GitHub https://github.com/emoro/mobile_food_environments[68].

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

## Acknowledgements

E.M. acknowledges support by Ministerio de Ciencia e Innovación/Agencia Española de Investigación (MCIN/AEI/10.13039/501100011033) through grant PID2019-106811GB-C32 and the National Science Foundation under Grant No. 2218748. K. de la H. acknowledges support from the University of Southern California Keck School of Medicine Dean's Pilot Funding Program, and the National Center for Advancing Translational Sciences under grant 1U54TR004279. B.M.B. acknowledges support from the Yale Cancer Prevention and Control Training Program, funded by the National Cancer Institute (T32 CA250803). A.L.H. acknowledges support from an NIH Ruth L Kirschstein National Research Service Award (NRSA) Institutional Training Grant (T32 5T32CA009492-35). The funders had no role in the study design, data collection and analysis, decision to publish, or preparation of the manuscript.

## Author contributions
B.G.-B., A.L.H., K. de la H. and E.M. designed research; B.G.-B. and E.M. performed research and analyzed the results. B.G.-B. and E.M. wrote the first draft of the manuscript. B.G.-B., A.L.H., B.M.B., M.B., B.B., A.P., K. de la H. and E.M. discussed the results and edited the manuscript. All authors approved the final version.

## Competing interests
The authors declare no competing interests.
