## [Peer Review File · Nature Communications]

REVIEWER COMMENTS

Reviewer #1 (Remarks to the Author):

Overall, the authors conducted quite solid work and the paper presents results that might be appealing to a broad audience. Below, are my comments and suggestions for improvement.

--- Originality and significance.

The paper uses fine-grained mobility data from mobile phones to study the eating habits of city dwellers from a data sample generated by almost ~750k people living in 11 US cities, for a period of 6 months. The motivation of the work is compelling: mobility is a crucial factor that influences eating habits, but previous studies have either only investigated food consumption in a static setting (e.g., availability of food venues in the area of residence) or they took mobility into account but were limited to small-scale investigations (mainly through surveys). The work fills this gap by providing the first picture of how mobility is associated with visits to fast-food outlets. The work is a significant and novel contribution because 1) it provides solid evidence that has the potential to change the mainstream perspective of the study of food consumption in urban settings; 2) it provides strong associations between visits to different types of food environments and a variety of measurable factors (sociodemographics, personal preferences, available of urban amenities); 3) it ventures beyond correlations to find causal relationships between the composition of food environments and the tendency to visit fast-food chains; 4) it does all of the above on one of the largest datasets that have been used for studying eating behavior. I anticipate that the contribution can spark broad interest across specialized communities (computational urbanism, computational social science, transportation & mobility) but also in the broader field of digital health and in the general public.

--- Key results.

The main working concept presented in the manuscript is that of mobile (built) food environments. Those represent the set of food outlets present in the urban areas where a person happens to be right before they visit a food venue. In this work, food environments are characterized mainly by the fraction of fast-food outlets over the total number of food outlets available in the area (as estimated from Foursquare data). The authors report three main findings:

- Mobile food environments are correlated with socioeconomic features that describe city dwellers
- A combination of mobile food environments and individual preferences can predict the frequency of fast food visits
- There is a causal relationship between food environments and the probability of visiting fast-food outlets

Last, the authors build upon these three findings to propose an intervention policy to limit the number of fast-food outlets in strategic areas such that visits to fast-foods are maximally reduced (compared to baseline strategies)

All results are interesting and novel, even though some aspects would need a deeper reflection (see Suggestions below).

--- Data, methodology, and conclusions.

The main data source is mobility data from Cuebiq. The authors combine it with foursquare and census data to characterize the urban spaces people visit and live in. The methodology is quite thorough and sound, although some of the choices sound a bit arbitrary (see Suggestions). Nevertheless, the authors provide several pieces of robustness analysis and manage to provide convincing arguments about their results being reliable under different conditions (e.g., filtering criteria). The methodology of causal inference that focuses on identifying individuals who changed their behavior over time is convincing. I have some doubts about the methods used to support the conclusions drawn from the simulation of intervention policies (see Suggestions below).

--- Clarity and context.

The writing is mostly quite clear. There are some minor clarifications that the authors should give in

the revised version, as highlighted below. The flow of the paper could be improved. At the moment, it suffers from the "traditional" structure of Nature-portfolio papers that force the writer to omit many methodological details and rush to the results. For example, all of the content in Methods could be presented earlier in a "just-in-time" fashion (e.g., describe the dataset when the data is introduced, spell out the full definition of food environment earlier on, etc.). When I read the paper I have to jump many times between the main text, methods, and supplementary material to get a good understanding of what the authors did. I am not sure what can be done to improve this if the current structure is enforced.

--- References

The references are appropriate and quite complete.

--- Suggestions (roughly sorted by importance)

- One of the most compelling points that the paper makes is that the mobile food environments correlate with both sociodemographic features of the visitors and with the likelihood of visiting fast food outlets much more than the home food environment. This finding supports the criticism that the authors direct towards previous research and the previously proposed "fast food bans" interventions focused on residential areas. In several passages, the authors seem to suggest that the home food environment does not matter much. I think that a more systematic reflection is needed on this aspect. Namely, I think the authors should clarify the reason behind the gap between mobile and home environments. Is the gap determined mainly by the concentration of fast food outlets in areas where people move to work and be entertained (rather than in residential areas), or because the food consumption behavior of people is inherently different at home and on the move? These two scenarios entail crucial implications: if the latter is the dominant reason, one could argue that it's not worth paying too much attention to fast food environments (food swamps) in residential areas. The dominant reason might be the former though: people are not attracted as much to fast foods around their home locations just because of the scarcer availability. The lower log-odds obtained for the DMV case in Figure 3A might indicate that the density of FFO is indeed a factor, as I imagine that the density of FFO is not particularly high around DMVs (the authors should check that). Also, I would recommend producing a second version of Figure 3A for home food environments only. In summary, a deeper reflection is needed around this crucial point, to avoid an overly rigid interpretation of the results that might lead to negatively (and unjustifiedly) affect home food environment studies in the future.

- The simulation experiments to measure the impact of hypothetical interventions are very nicely thought out. The reported results strictly depend on the model that predicts fast food visits from the combination of the context and user profiles. Therefore, the reliability and validity of the results of different interventions are strictly dependent on the accuracy of that predictive model. I don't think there's any mention of the error/accuracy of the model anywhere in the paper or in the supplementary materials. I don't think that the accuracy of the model impacts in any way the results presented in the first part of the paper -- the high coefficients indicate that there are strong statistical associations. However, accuracy becomes crucial when the model is used to cast predictions (in this case, a prediction of different fast-food ban scenarios). The authors should evaluate the predictive performance of the model. If the predictive power is low, then the section about possible interventions should be revised in depth. I'd very much welcome the authors' thoughts on this matter.

- The terminology used to define food environments might introduce some confusion or ambiguity in the semantics of some experiments. I wonder if I have interpreted the manuscript correctly in this respect. Given the definition of food environment $\phi(x)$ from line 332, a food environment can be calculated on *any* stop location of a given user. Referring to Figure 1A, one could extract 5 food environments, one for each of the locations the user stops at. But I believe that the only meaningful

food environments to study are the home food environment and the "context" (respectively, the two dashed circles in Figure 1A). I wonder if this is always the case though. Are the first two experiments that the authors report --correlation between food environment with i) socioeconomic and ii) FFO visits-- always conducted on either referred to the home location (to model the home food environment) and the "context" (to model the mobile food environment)? If that's the case, it would be good to clearly spell it out and perhaps define "context" earlier on ("context" is mentioned at the beginning but it is defined more in detail starting only from line 139). If that's not the case, then the authors would need to explain why they considered all stop locations, as I don't think that would be ideal.

- In line 143, the authors write: "For our regression, we only consider users that went to both fast and non-fast food outlets at lunch at least once during the 6-month observation period" -> this choice could be questioned, as it causes the removal of 20%+ of users who never visit FFO. These users effectively take part in the same process of mobility and food consumption, so I don't think that excluding them is a sensible choice. I'd be interested to hear the author's opinion about it.

- The authors consider food visits only those visits at food venues that last from 5 minutes to 2 hours. I wonder why the temporal limitation is needed. Why can't one just consider venue type, regardless of the time spent? How many visits are discarded with this type of temporal filter? Showing the distribution of stop times would be informative

- I wonder why the authors georeferenced the users at the level of Census Block Groups and not at the level of Census Blocks. Is it because of the lack of data at the level of blocks?

- In the introduction, the authors write: "low-quality food environments are frequently concentrated among low-income communities and communities of color". This statement needs to be contextualized: this is mostly true for the US, not necessarily elsewhere. Indeed, the vast majority of studies covered by the two survey papers cited by the authors have been conducted in the US. Authors should specify the geographical context for these claims. I believe that "in the US" should be part of the paper title. Also, the authors should avoid using the word "Americans" as a synonym for the US/Canadian residents.

- The abstract could be tweaked slightly to make the results clearer for a reader that hasn't read the rest of the paper. "increases the odds of people visiting an FFO by 20%" -> it's not clear who these people are; one could think that these are people living in areas close to the FFOs. "10% more FFO in an area" -> 10% more than what, exactly?. While point i) should be rather clear to anyone, points ii) and iii) might be not very clear for readers who are not familiar with the details of the methodology (I did not fully understand them when I read them first).

- Why is the line in Fig 2b discontinued between 10 pm and midnight? Why hasn't the data from those temporal windows been considered?

- The result reported in line 92 (Fig2) lacks interpretation. Why is public transport negatively correlated with visits to FFO? Is it perhaps because the use of public transport is partly (inversely) co-linear with the length of commuting? Or maybe because there is an association between the type of transport and the type of job people have?

- In line 92, it seems that "low-skill jobs" should be "high-skill jobs" instead (based on the results in Fig2)

- Line 158 "significant portion of user" -> it would be more correct to say "small fraction of users" (it's <1%)

- The sentence starting on line 179 is a bit convoluted and hard to parse

- typo in the abstract: diet and related disease -> diet and related diseases

- format issues with references e.g, line 45: "For instance, in⁹ the" -> using references as nouns makes it hard to parse the sentence, because of the specific reference format. Authors could expand them e.g., as follows: "For instance, in a study by X et al..."

Reviewer #2 (Remarks to the Author):

The authors present a very interesting paper on the relevance of the mobile food environment for policy interventions. In particular, they focus on fast-food visits, quantifying the correlation with socio-demographic and mobility traits, and suggesting a strategy called "Behavior-Environment intervention" to prioritize points of interest and reduce the number of visits to fast-foods. To do so, they leverage mobility data capturing 1.86 million anonymized users over 11 US metropolitan areas.

For the work to be of significance to public health more in-depth analysis is needed. Specifically, the manuscript currently misses convincing "data stories", translating the model potential into tangible results to be discussed by a broader audience. This translational effort is often what makes the difference between articles published in Nature Communications, and more technical journals.

My specific comments:

- 1) While the data covers 11 US metropolitan areas, in the manuscript we find only minimal comments on LA county. What have you learned by applying the behavior-environment intervention to different metropolitan areas? Could you stratify your results and summarize them? Which are the metropolitan areas with more distinctive results? Please discuss some examples (in line with Table S1).
- 2) The ratio between the number of fast food outlets and the total number of food outlets within a given radius is one of the many possible ways to capture the prevalence of fast-foods in a certain area. However, this ratio does not imply "enrichment". Once set the spatial unit corresponding to the authority that regulates the number of fast-foods in a given area (e.g., county), it is worth identifying which subunits are enriched in fast-foods (hypergeometric test).
- 3) To properly comment Figure 1B you need to provide an effect size estimation and a p-value. Please, make sure to avoid "overpowering".
- 4) Lines 95-96. Could you elaborate on how a higher percentage of FFO occurs in neighborhoods with higher levels of educational attainments?
- 5) Lines 103-104. μ_i captures a heterogeneous measure across individuals and as such, you should avoid reporting it with average and standard deviation. Moreover, this is a quite important measure whose distribution should be plotted and discussed, as it influences the following logistic models and intervention strategies. Ideally, the results should be commented under the lens of extreme value theory.
- 6) Lines 89-117 and SI Section 7.1. Before drawing conclusions on the R^2 provided by a simple linear model, the authors should consider applying GLM with proper data transformation. Even other machine-learning-based non-linear regressors should be investigated (e.g., random forests).
- 7) Given the high granularity of the data characterizing food outlets (SI Section 3), the authors should consider stratifying their results and intervention for specific fast-food chains. If this is challenging, the authors should clearly discuss the reason.
- 8) What is the degree of universality of the results presented for fast-foods? In other words, if the focus is on different food outlets, what do you expect to observe? Please expand the discussion accordingly.
- 9) Line 223. Converting a FFO into a non-FFO could work theoretically but it is worth exploring how this simple intervention translates to reality. According to what you have learned about human behavior and mobility, are you able to come up with more practical suggestions? Is it possible to identify which non-FFO is ideal for a given location?

Reviewer #3 (Remarks to the Author):

The manuscript seeks to address an important research need in food environment research, addressing restaurants beyond commonly used static models. The use of mobile data is innovative. Yet, due to this, the explanation of the methods and approach, while comprehensive, missed some important details to evaluate the information presented.

Information about the participants was lacking or not clear. While the data were anonymous, it would have helped to know the overall demographics and key aspects of the people providing the data. Some aspects include age, gender, and employment status, at minimum, as factors that would influence the information collected.

Some aspects of the analysis were not well justified. For example the DMV experiment visit. In practice, I wonder about the choice, given that people may eat before or after, not really having to eat in the area (as opposed to a place of employment).

Some aspects of the policy simulation needed additional clarification. For example, the discussion of changing FFO to non-FFO: what are these non-FFO? Are these restaurants? Or other establishments?

Reviewer #4 (Remarks to the Author):

This paper presents an analysis of the effect of mobile food environments on fast food visits. The analysis is based on individual-level anonymized mobility data collected in 11 US metropolitan areas and provided by Cuebiq over a period of 6 months (from October 2016 and March 2017). The authors propose to identify for each user a home and mobile food environments (based on individual mobility information and locations of Food Outlets (FO) and Fast Food Outlets (FFO) identify with Foursquares). They also (not very clearly) identify individual FO and FFO visits. They finally use these information to assess how food choices may be influenced by food environments. The paper is well-written and presents an interesting piece of work. However, I am not convinced that this work should be published in a high impact journal like Nature Communications. I listed below three major concerns regarding this paper.

1) First, the study is far from being reproducible, the data are only available upon request submitted to Cuebiq. I followed the link provided by the authors, and, after several attempts, it seems that it will be quite complicated and time consuming to get similar Cuebiq/spectus data to that used by the authors to perform the analysis. The other data used (American Community Survey and PLACES Local Data) should also be clearly identified. Finally, the link provided by the authors to reach the code used to run the analysis is not available and there is no mobile_food_environments repository on emoro GitHub account... In my opinion, the paper does not meet the current data (and code) sharing standard for publication. I can therefore not recommend this paper for publication without a permanent link to access the data used to perform the analysis.

2) Second, I would recommend the authors to give more details about the data and the cleaning process (algorithms and parameters used to identify the home locations, FO/FFO visits...). I maybe missed something but it is not clear to me how the authors have identified the FO/FFO visits. Is it based on a check-in or individual presence in the building? Most of the findings are based on these "visits", I would encourage the authors to give more details regarding the methodology used to identify these visits and to perform a thorough analysis of FO and FFO daily visits at an individual level (number of FO or FFO visits per user, day and census track, identification and validation of individual routines...).

3) Finally, I would recommend the authors to add a control mechanism to handle potential problems of spatial autocorrelation. Indeed, spatial context and FO/FFO visits can be spatially linked if the distance

between context and visit is too small (as shown in Table S3) . It would recommend the authors to used an increasing d_{min} instead of a d_{max} to clearly assess the effect of spatial correlation.

Response to the reviewers

We thank the reviewers for their supportive, critical and extensive assessment of our work “You are where you eat: Effect of mobile food environments on fast food visits” submitted for consideration for publication in Nature Communications (reference #NCOMMS-22-32906A-Z).

In the following, we address their concerns point by point. We also specify the changes made to the main paper and the Supplementary Material to address them.

Reviewer 1

Reviewer Point P 1.1 — Overall, the authors conducted quite solid work and the paper presents results that might be appealing to a broad audience. Below, are my comments and suggestions for improvement.

Reply: We greatly appreciate your constructive and positive comments and suggestions.

Reviewer Point P 1.2 — Originality and significance.

The paper uses fine-grained mobility data from mobile phones to study the eating habits of city dwellers from a data sample generated by almost 750k people living in 11 US cities, for a period of 6 months. The motivation of the work is compelling: mobility is a crucial factor that influences eating habits, but previous studies have either only investigated food consumption in a static setting (e.g., availability of food venues in the area of residence) or they took mobility into account but were limited to small-scale investigations (mainly through surveys). The work fills this gap by providing the first picture of how mobility is associated with visits to fast-food outlets. The work is a significant and novel contribution because 1) it provides solid evidence that has the potential to change the mainstream perspective of the study of food consumption in urban settings; 2) it provides strong associations between visits to different types of food environments and a variety of measurable factors (sociodemographics, personal preferences, available of urban amenities); 3) it ventures beyond correlations to find causal relationships between the composition of food environments and the tendency to visit fast-food chains; 4) it does all of the above on one of the largest datasets that have been used for studying eating behavior. I anticipate that the contribution can spark broad interest across specialized communities (computational urbanism, computational social science, transportation & mobility) but also in the broader field of digital health and in the general public.

Reply: We greatly appreciate your enthusiasm about our results and contribution to the field.

Reviewer Point P 1.3 — Key results.

The main working concept presented in the manuscript is that of mobile (built) food environments. Those represent the set of food outlets present in the urban areas where a person happens to be right before they visit a food venue. In this work, food environments are characterized mainly by the fraction of fast-food outlets over the total number of food outlets available in the area (as estimated from Foursquare data). The authors report three main findings: - Mobile food environments are correlated with socioeconomic features that describe city dwellers - A combination of mobile food environments and individual preferences can predict the frequency of fast food visits - There is a

causal relationship between food environments and the probability of visiting fast-food outlets. Last, the authors build upon these three findings to propose an intervention policy to limit the number of fast-food outlets in strategic areas such that visits to fast-foods are maximally reduced (compared to baseline strategies). All results are interesting and novel, even though some aspects would need a deeper reflection (see Suggestions below).

Reply: We appreciate your positive comments and suggestions, which we address below.

Reviewer Point P 1.4 — Data, methodology, and conclusions.

The main data source is mobility data from Cuebiq. The authors combine it with foursquare and census data to characterize the urban spaces people visit and live in. The methodology is quite thorough and sound, although some of the choices sound a bit arbitrary (see Suggestions). Nevertheless, the authors provide several pieces of robustness analysis and manage to provide convincing arguments about their results being reliable under different conditions (e.g., filtering criteria). The methodology of causal inference that focuses on identifying individuals who changed their behavior over time is convincing. I have some doubts about the methods used to support the conclusions drawn from the simulation of intervention policies (see Suggestions below).

Reply: We appreciate your positive comments and suggestions, which we address below.

Reviewer Point P 1.5 — Clarity and context.

The writing is mostly quite clear. There are some minor clarifications that the authors should give in the revised version, as highlighted below. The flow of the paper could be improved. At the moment, it suffers from the "traditional" structure of Nature-portfolio papers that force the writer to omit many methodological details and rush to the results. For example, all of the content in Methods could be presented earlier in a "just-in-time" fashion (e.g., describe the dataset when the data is introduced, spell out the full definition of food environment earlier on, etc.). When I read the paper I have to jump many times between the main text, methods, and supplementary material to get a good understanding of what the authors did. I am not sure what can be done to improve this if the current structure is enforced.

Reply: We thank the reviewer for the suggestions and agree that both ways to present (just-in-time fashion) or as it now has advantages and disadvantages. The current structure has the advantage that it can communicate the results in a way that's friendly to technical and less-technical audiences. However, as the reviewer points out, it can leave more prepared readers with the burden of jumping back and forth.

We think that a way to harness the advantages of the current format while partially relieving the burden to the audience jumping back and forth is to expand the introductory paragraphs of the studies to include some more relevant data and methodological details. To this end, we revised the text and added a summarizing introduction to the second and third studies (moving people study, RMV study) to be a bit clearer.

Reviewer Point P 1.6 — References

The references are appropriate and quite complete.

Reply: We appreciate your positive comment about our references.

Reviewer Point P 1.7 — Suggestions (roughly sorted by importance)

- One of the most compelling points that the paper makes is that the mobile food environments correlate with both sociodemographic features of the visitors and with the likelihood of visiting fast food outlets much more than the home food environment. This finding supports the criticism that the authors direct towards previous research and the previously proposed "fast food bans" interventions focused on residential areas. In several passages, the authors seem to suggest that the home food environment does not matter much. I think that a more systematic reflection is needed on this aspect. Namely, I think the authors should clarify the reason behind the gap between mobile and home environments. Is the gap determined mainly by the concentration of fast food outlets in areas where people move to work and be entertained (rather than in residential areas), or because the food consumption behavior of people is inherently different at home and on the move? These two scenarios entail crucial implications: if the latter is the dominant reason, one could argue that it's not worth paying too much attention to fast food environments (food swamps) in residential areas. The dominant reason might be the former though: people are not attracted as much to fast foods around their home locations just because of the scarcer availability. The lower log-odds obtained for the DMV case in Figure 3A might indicate that the density of FFO is indeed a factor, as I imagine that the density of FFO is not particularly high around DMVs (the authors should check that). Also, I would recommend producing a second version of Figure 3A for home food environments only. In summary, a deeper reflection is needed around this crucial point, to avoid an overly rigid interpretation of the results that might lead to negatively (and unjustifiedly) affect home food environment studies in the future.

Reply: We thank the reviewer for raising this important point. Our results do indeed show that home food environments are less important to understand overall visits to FFO. But the main reason is that **the majority of food visits happen very far away from home**. Thus, all else being equal between mobile and home environments and behaviors occurring in those environments, it is probabilistically more likely that more FFO visits would happen in mobile environments. As we say in the *Characterization of mobile food environments* section

The median distance to any type of FO visited is 6.94km (Interquartile Range, IQR, [2.30km - 17.23km]), but this varies by outlet type: the median distance to grocery stores/supermarkets is much smaller, 3.1km (IQR [1.35km - 8.22km]), while FFO is 6.74km (IQR [2.50km-16.62km]) away (median). In fact, only 6.8% of the visits to FFO occur within a user's home census tract. Thus, most fast food visits occur in food environments outside of a user's home neighborhood.

We argue from the outset of our Results section that home environments are thus less significant in comprehending the behavior of users when it comes to fast-food visits. Not only that, we also show and comment at the beginning of the Results section that home and mobile food environments are very different in the exposure to FFO [$\rho(\phi_i^m, \phi_i^h) = 0.191 \pm 0.002$]. Thus the most important environments to understand visits to FFO are the mobile food environments because they are the most frequented ones when visiting food outlets.

The reviewer also points to other possibilities which are not related to the frequency of food outings in different environments, but to the different characteristics of those environments: i) One is that the concentration of fast food outlets is different between home and mobile food environments. In fact, as the reviewer suggests, we already studied that possibility by showing that mobile

food environments have more ratio of FFO to FO than home environments: "the time-weighted ratio of FFO to FO that a user is exposed to [...] (ϕ_i^m see Methods) has a median of 14.1%", while "home environments have a relative low FFO to FO ratio [median of ϕ_i^h for all users is 9.75%]".

ii) The second possibility is that users behave very differently at home and on the move. This is an interesting possibility. The reviewer suggests to repeat figure 3A only for home environments. Note, however, that our model Eq. (1) cannot be restricted only home food environments, since it is based on identifying the response of individuals to different contexts. Thus, for actions around home we get $\phi(c_{it}) \simeq \phi(\mathbf{h}_i)$ always and there would be no variability in the regressors to train the model. Also, the majority of contexts before lunch happen away from home. Nevertheless, we have studied this second possibility by analyzing the actions that happen closer to home environments. The results are presented in Supplementary Note 7.2, where we can see that the impact of the contexts close to homes is slightly smaller so, as the reviewer suggests, people slightly have different consumption behaviors at home and on the move. This also happens in other particular places. For example, we find that food environments are not less dense in FFO around DMVs (median of $\phi(c_{it}) = 0.142$) than around all lunch contexts (median of $\phi(c_{it}) = 0.137$). So the small difference in the log-odds is due to different responses of users to the mobile food environment around those areas.

However, as we said before, despite those minor differences between the environments and behaviors within them, the main reason why mobile food environments are more important than home environments and, in turn, they correlated more with FFO visits, is that most food outings happen far away from home. We have commented on those slight differences mentioned by the reviewer and slightly modified the wording in the Results section to clarify this further.

Changes to the text: We have modified the text in the Results section to stress that it is the frequency of food outings away from home that makes mobile food environments more important to understand visits to fast-food outlets. We have also included the analysis as a function of the distance to home in the Supplementary Note 7.2.

Reviewer Point P 1.8 — The simulation experiments to measure the impact of hypothetical interventions are very nicely thought out. The reported results strictly depend on the model that predicts fast food visits from the combination of the context and user profiles. Therefore, the reliability and validity of the results of different interventions are strictly dependent on the accuracy of that predictive model. I don't think there's any mention of the error/accuracy of the model anywhere in the paper or in the supplementary materials. I don't think that the accuracy of the model impacts in any way the results presented in the first part of the paper – the high coefficients indicate that there are strong statistical associations. However, accuracy becomes crucial when the model is used to cast predictions (in this case, a prediction of different fast-food ban scenarios). The authors should evaluate the predictive performance of the model. If the predictive power is low, then the section about possible interventions should be revised in depth. I'd very much welcome the authors' thoughts on this matter.

Reply: We thank the reviewer for this critical point. Although some performance metrics were included in the previous version of the Supplementary Material, we recognize that the point raised by the reviewers merits a more comprehensive study of the predicting power of the model used in our intervention. For that reason, we have extended the Supplementary Note 7 to investigate the individual and area performance of our logistic regression model using different out-of-sample

strategies. In particular, we have used a “geographical” out-of-sample strategy in which the model is trained in the food outings of users in a particular set of areas in the city and tested in the rest of the areas. This way, we test the performance of the model when users are subject to unseen or different environments, a situation similar to our interventions. Although the model shows large accuracy (around 80%) for the test sets at the individual level, it is more precise at the area level with correlations of 95% or 70% (both in the test set) for the total number of visits to FFO or the fraction of FFO visits to food visits in different areas. These results show that our model is suitable for predicting FFO visits under different fast-food interventions.

Changes to the text: We have expanded SM Section 7 to include results on the model’s predictive accuracy for several performance metrics. We have also commented on the performance of our model in the Methods section of the main paper.

Reviewer Point P 1.9 — The terminology used to define food environments might introduce some confusion or ambiguity in the semantics of some experiments. I wonder if I have interpreted the manuscript correctly in this respect. Given the definition of food environment $\phi(x)$ from line 332, a food environment can be calculated on *any* stop location of a given user. Referring to Figure 1A, one could extract 5 food environments, one for each of the locations the user stops at. But I believe that the only meaningful food environments to study are the home food environment and the “context” (respectively, the two dashed circles in Figure 1A). I wonder if this is always the case though. Are the first two experiments that the authors report –correlation between food environment with i) socioeconomic and ii) FFO visits– always conducted on either referred to the home location (to model the home food environment) and the “context” (to model the mobile food environment)? If that’s the case, it would be good to clearly spell it out and perhaps define “context” earlier on (“context” is mentioned at the beginning but it is defined more in detail starting only from line 139). If that’s not the case, then the authors would need to explain why they considered all stop locations, as I don’t think that would be ideal.

Reply: Thank you for pointing out the potential misinterpretation between environment and context due to a lack of clarity in our writing. As the reviewer correctly states, for every given place x we can define the mobile food environment around it and characterize it by $\phi(x)$. Context is a particular food environment: the one the user was in just before going for lunch. We have made changes to the main text and Methods to stress this difference between general environments and contexts.

In our paper, we use $\phi(x)$ in the first part of our results to show how different (in terms of fast-food) people’s home neighborhoods are from the areas they are exposed to throughout the day. Also to compare with other literature that uses a similar description of fast-food exposure [4, 2, 3]. In the second part, to be more quantitative we restrict that exposure to the context, i.e., the environment just before lunch.

An important question is whether contexts are different from the rest of the environments. That is, it could be that people are not surrounded by FFO throughout the day, but they are just before lunch. This is not the case as we find a (Pearson’s) correlation of 0.662 ± 0.001 between the ϕ_i^m calculated using all stops and only contexts before food outings. Thus, even ϕ_i^m calculated using all stops is also a good description of the contexts before food outings.

Changes to the text: We have explicitly defined “context” in the text before equation (1) to be more precise. Also, the word “context” is only mentioned when it is introduced to avoid potential misinterpretations. We have also modified the methods section to introduce and differentiate the contexts from the rest of the environments.

Reviewer Point P 1.10 — In line 143, the authors write: “For our regression, we only consider users that went to both fast and non-fast food outlets at lunch at least once during the 6-month observation period”. This choice could be questioned, as it causes the removal of 20%+ of users who never visit FFO. These users effectively take part in the same process of mobility and food consumption, so I don’t think that excluding them is a sensible choice. I’d be interested to hear the author’s opinion about it.

Reply: The exclusion of those users is only a technical requirement to fit the logistic regression model. Logistic regressions with fixed factors per individual (α_i) can only be fitted numerically if those individuals have different observations y_{it} . Theoretically speaking those users are included in the regression. But since their choice (FFO or non-FFO) is always the same independently of the context, their fixed effect α_i will be $\pm\infty$. Thus, their inclusion would not modify our results for model (1) and thus they are effectively considered in our analysis.

Furthermore, since those users are not affected by the environment, they would not change their behavior under different interventions. So our results in the *Policy implications* section would not be modified either.

Reviewer Point P 1.11 — The authors consider food visits only those visits at food venues that last from 5 minutes to 2 hours. I wonder why the temporal limitation is needed. Why can’t one just consider venue type, regardless of the time spent? How many visits are discarded with this type of temporal filter? Showing the distribution of stop times would be informative

Reply: We only consider stays that last more than 5 minutes for several reasons: first and most importantly, shorter stays cannot be detected with enough precision by our algorithm [10]. Furthermore, we want to discard people just passing by or stopping for a short time close to the food place. On the other hand, we consider only stays shorter than 2 hours to discard stays related to working places. That is, because some of our users are not visiting the food outlet but rather working there. With this threshold, we discarded only 5.83% of the visits to food outlets.

We have included more information about how stays and visits are detected, including the distribution of stopping times in the Supplementary Note 1. As we can see there, we are only discarding the most unfrequent durations of the stops.

Changes to the text: We have expanded the Supplementary Note 1 to give more details about how stays and visits are detected. We have also given more descriptive information about our data in that section, including the distribution of stop times.

Reviewer Point P 1.12 — I wonder why the authors georeferenced the users at the level of Census Block Groups and not at the level of Census Blocks. Is it because of the lack of data at the level of blocks?

Reply: Thanks for the comment. Although we could georeference users at the Census Block, we used Census Block Groups for two reasons: i) first and, more importantly, for privacy reasons.

Larger areas have more population and make our data more privacy-preserving. ii) Secondly because important demographic characteristics of the census areas like median income, racial composition and transportation use are only available in the American Community Survey (ACS) at the Census Block Group, see [16].

Reviewer Point P 1.13 — In the introduction, the authors write: "low-quality food environments are frequently concentrated among low-income communities and communities of color". This statement needs to be contextualized: this is mostly true for the US, not necessarily elsewhere. Indeed, the vast majority of studies covered by the two survey papers cited by the authors have been conducted in the US. Authors should specify the geographical context for these claims. I believe that "in the US" should be part of the paper title. Also, the authors should avoid using the word "Americans" as a synonym for the US/Canadian residents.

Reply: Thanks a lot for the suggestion. Indeed, the results were obtained from the US so we agree that it's good to specify this. We have updated the abstract and body of the article to be explicit on that the results were obtained with US data.

Changes to the text: We have modified the abstract and body of the article to specify the US context of our results.

Reviewer Point P 1.14 — The abstract could be tweaked slightly to make the results clearer for a reader that hasn't read the rest of the paper. "increases the odds of people visiting an FFO by 20%" → it's not clear who these people are; one could think that these are people living in areas close to the FFOs. "10% more FFO in an area" → 10% more than what, exactly?. While point i) should be rather clear to anyone, points ii) and iii) might be not very clear for readers who are not familiar with the details of the methodology (I did not fully understand them when I read them first).

Reply: Thanks for the note; we reviewed the abstract to read: "Using a semi-causal framework and various natural experiments, we find that a 10% higher prevalence of FFO (across all food outlets) in an area increases the odds of people moving within it to visit a FFO by approximately 20%.

Reviewer Point P 1.15 — Why is the line in Fig 2b discontinued between 10 pm and midnight? Why hasn't the data from those temporal windows been considered?

Reply: Previous Fig 2b was created using one panel per day. The apparent discontinuity was an artifact of the spaces around each panel. We have produced a new continuous figure throughout the week to resolve this issue. No data was and is discarded for any temporal window in our analysis.

Changes to the text: We have modified Fig 2b to resolve the visualization issue created by having one panel per day.

Reviewer Point P 1.16 — The result reported in line 92 (Fig2) lacks interpretation. Why is public transport negatively correlated with visits to FFO? Is it perhaps because the use of public transport is partly (inversely) co-linear with the length of commuting? Or maybe because there is an association between the type of transport and the type of job people have?

Reply: We thank the reviewer for pointing out this important finding. Note that our models are multivariate. Thus the correct interpretation of the results in Figure 2 in the main paper is that the coefficient estimate for each demographic variable measures the association of that demographic variable conditioned on the rest of the variables kept constant. However, demographic variables are not independent, and that cannot be done without affecting the variable itself. As we see in Figure S5 in the Supplementary Material, demographic variables like income, education, employment, or low-skill jobs for each CBG are moderately correlated. In any case, in Figure S5 in the Supplementary Material, we also present results for a single univariate model $\phi_i^m, \phi_i^h, \mu_i \sim \beta_l d_{l,i} + MSA_i$ where, as mentioned by the reviewer, we can still see that public transport is negatively correlated with visits to FFO.

Note that, in the census data, the use of public transportation is correlated (Pearson's correlation $\rho = 0.357 \pm 0.001$, see Figure S5) with the length of commuting. Thus, people that take public transportation tend to have longer commuting (note that the length of commuting is measured in time, not distance).

The explanation of why public transport is negatively correlated with visits to FFO is simple: as we can see in Figure 2A (and Figure S5), the use of more public transportation is associated with mobile food environments which have less FFO. Thus, those users are less exposed to FFO as they move through the day. It is well-established that lack of access to a vehicle, which is often followed by increased use of public transport, can inhibit food access and impact food visits [6]. Few studies exist exploring modes of transport to restaurants and FFO specifically. A study of Detroit residents found that survey respondents reporting more frequent visits to FFO were more likely to own and use a car than to access the restaurants via public transport; only a small fraction of respondents reported using public transport to travel to restaurants [7]. Similarly, USDA Economic Research Service analysis of evidence from the National Food Acquisition and Purchase Survey (April 2012-January 2013) has shown that food insecure populations, who are generally of lower-income, are more likely to walk, bike, or take public transit to access grocery stores than higher-income food secure populations [17].

In any case, the coefficient in Fig 2A is small (-0.106 ± 0.0015). Thus the effect of using public transportation is very small on fast food visits.

Changes to the text: We have added some interpretation about this finding in the main text

Reviewer Point P 1.17 — In line 92, it seems that "low-skill jobs" should be "high-skill jobs" instead (based on the results in Fig2)

Reply: Thanks a lot for realizing this. This was indeed a typo. It has been corrected.

Reviewer Point P 1.18 — Line 158 "significant portion of user" → it would be more correct to say "small fraction of users" (it's < 1%)

Reply: Thanks a lot for the correction. We have changed it to:

"Using the time series of the different contexts before lunch, we were able to detect a small fraction of users (0.43%) that changed their habitual context before lunch within our observation period (see Methods and Supplementary Note 5 for details about this detection). Due to our large sample size, this small fraction had a large enough size (approx 8k users) to do statistical inference."

Reviewer Point P 1.19 — The sentence starting on line 179 is a bit convoluted and hard to parse

Reply: Thanks a lot for noting this. We have now changed the text to read:

"Finally, we sought to analyze a setting as close as possible to one where people are placed in random places within the city. Studying food decisions under such circumstances would preclude potential estimation bias from omitted variables affecting both food preference and location in the city. For this we propose a natural experiment looking at people who visited DMV/RMV, the food environment they faced and the decision they took to get a meal. In particular we repeat the analysis made for Eq. (1) but only when the context before lunch is the Department of Motor Vehicles (DMVs)."

Reviewer Point P 1.20 — typo in the abstract: diet and related disease → diet and related diseases

Reply: Thanks a lot for detecting this. We have now corrected it.

Reviewer Point P 1.21 — Format issues with references e.g, line 45: "For instance, in [9] the" → using references as nouns makes it hard to parse the sentence, because of the specific reference format. Authors could expand them e.g., as follows: "For instance, in a study by X et al..."

Reply: Thanks a lot for noting this. We corrected this as well as another instance using references as nouns.

Reviewer 2

Reviewer Point P 2.1 — The authors present a very interesting paper on the relevance of the mobile food environment for policy interventions. In particular, they focus on fast-food visits, quantifying the correlation with socio-demographic and mobility traits, and suggesting a strategy called “Behavior-Environment intervention” to prioritize points of interest and reduce the number of visits to fast-foods. To do so, they leverage mobility data capturing 1.86 million anonymized users over 11 US metropolitan areas.

For the work to be of significance to public health more in-depth analysis is needed. Specifically, the manuscript currently misses convincing “data stories”, translating the model potential into tangible results to be discussed by a broader audience. This translational effort is often what makes the difference between articles published in Nature Communications, and more technical journals.

Reply: We thank the reviewer for the thoughtful suggestion. Indeed the translation of the results and implications of our research for a broader audience might increase the likelihood that policymakers and stakeholders can benefit from the analysis done here. In this sense, our paper contains a clear message for the broad audience (see, e.g., the abstract or our conclusions): our behavioral results would allow designing interventions that are more powerful (up to 4x) than other interventions that focus on just the environmental features, as is commonly done by using USDA metrics to identify food swamps or deserts. We believe this is a powerful data story because it changes the focus from static areas of intervention to dynamic behavioral places. The fact that mobility data is increasingly available for researchers and policymakers will help to identify better each community’s “behavioral food hot spot”, where it happens, and tailor future environment “smart cities” interventions there.

Furthermore, our manuscript contains a long *Policy implications* section in which we provide a lot of information about how those interventions would look like (through area interventions, apps, etc.) and where they could happen. Our analysis at the end of the Policy Interventions section shows that areas which are typically under-considered in interventions, like Airports, Factories, or Offices, could be more efficient to intervene than places in restaurant or entertainment areas. Also, in our conclusions, we mention how our results can be translated into tangible and broader results across different groups and other types of food, making them more efficient, holistic, and equitable.

We believe all this information makes our results tangible and ready to be used by policymakers. Several billions of dollars are spent every year in the US on healthy food initiatives using static environments. We believe our results show an important data story: complement those initiatives with behavioral data to understand how, when, and where people consume less healthy food.

Changes to the text: We have considerably changed the abstract, the introduction, our results, and the conclusions to discuss different ways in which our results could be translated into tangible results. For example, we added a whole paragraph in the Policy Implications section to show how our policy example could be implemented in reality. We also modified the Conclusions to

show how our models could be used for more holistic interventions across different types of food, different groups, and geographies.

Reviewer Point P 2.2 — While the data covers 11 US metropolitan areas, in the manuscript we find only minimal comments on LA county. What have you learned by applying the behavior-environment intervention to different metropolitan areas? Could you stratify your results and summarize them? Which are the metropolitan areas with more distinctive results? Please discuss some examples (in line with Table S1).

Reply: The objective of the paper is to study the effect of mobile food environments on FFO visits in large urban areas. Thus, our analysis is done by putting together all FFO visits in those large urban areas. For that reason, in our paper, no mention is done of any particular geography, apart from some mention of literature on fast food bans in LA (our reference [14] and the use of the LA map in Figure 1 to help the reader understand our results.

The reviewer is correct in pointing out, though, that our results might vary across geographical areas. Thus, it is important to quantify if our results still hold for different cities. To test that possibility, we have studied the main results of the paper (Figure 2 and Figure 4) by city. Results are presented in the new Supplementary Note 10, where we can see that our main results still hold by city. We also see a small difference in the relative size of different interventions by city. The study of these small differences could be a future avenue for a specific more qualitative study that is beyond the scope of our objective in this paper. Nevertheless, we have commented on those differences in the Supplementary material and referred to the fact our main results in the body of the paper hold for individual cities.

Changes to the text: We have included a new Supplementary Note 10 to investigate the potential difference of our results by city and changed the main text to refer to this analysis.

Reviewer Point P 2.3 — The ratio between the number of fast food outlets and the total number of food outlets within a given radius is one of the many possible ways to capture the prevalence of fast-foods in a certain area. However, this ratio does not imply “enrichment”. Once set the spatial unit corresponding to the authority that regulates the number of fast-foods in a given area (e.g., county), it is worth identifying which subunits are enriched in fast-foods (hypergeometric test).

Reply: We thank the reviewer for this comment. Note that our analysis and models include a fixed factor by MSA (city) or a fixed factor by individual (people living in a given city), and thus the effect of the environment $\phi(x)$ in our models is calculated relative to the city. So we are taking into account the “enrichment” of an area or context to the city. As suggested by the reviewer, we have also tested the explicit use of fixed factors by city and county in model (1) to test the potential effect of $\phi(x)$ relative to the area, rather than the absolute ratio of fast-food in a certain area. This robustness check is now included in the Supplementary Note 12.3, where we have tested the following regression model:

$$\Pr(y_{it} = 1) = \text{logit}^{-1}[\beta_0 + \alpha_i + \delta_t + \beta\phi(\mathbf{c}_{it}) + A_{it}] \quad (1)$$

where A_{it} is the county or the MSA where the decision is made. Our results show that the effect of the context (the β coefficient in the above model) is largely independent of having county or MSA fixed factors. Thus, users seem to be affected directly by the ratio of FFO in an area, not the

relative "enrichment" of fast-food options in that area to the city or the county.

Changes to the text: We have included a new robustness check in Supplementary Note 12.3 to study the FFO "enrichment" of an area relative to the county or the city.

Reviewer Point P 2.4 — To properly comment Figure 1B you need to provide an effect size estimation and a p-value. Please, make sure to avoid "overpowering".

Reply: Figure 1B is provided to show the contrast between the distance traveled to different venues. Note that we do not statistically compare the distributions at any point in the paper. In the text, we only give the medians of those distributions to illustrate the different distances traveled to all venues, food venues, or grocery stores. To make that comparison more quantitative, we have included the interquartile ranges for those medians. Note that both medians and those interquartile ranges are not affected by overpowering.

Reviewer Point P 2.5 — Lines 95-96. Could you elaborate on how a higher percentage of FFO occurs in neighborhoods with higher levels of educational attainments?

Reply: Note that our models are multivariate. Thus the correct interpretation of the results in Figure 2 in the main paper is that the coefficient estimate for each demographic variable measures the association of that demographic variable conditioned on the rest of the variables kept constant. However, demographic variables are not independent and that cannot be done without affecting the variable itself. As we see in Figure S5 in the Supplementary Material demographic variables like income, education, employment or low-skill jobs for each CBG are moderately correlated. For that reason, results for a single univariate model like $\phi_i^m, \phi_i^h, \mu_i \sim \beta_l d_{l,i} + MSA_i$ are different for some of those variables, see Figure S5 in the Supplementary Material. Especially significant are cases like the relationship between educational attainment and the fraction of FFO in home neighborhoods, ϕ_i^h . As we can see the univariate relationship is negative (more educational attainment in the neighborhood is related to less FFO in that neighborhood), as expected. However, when we perform the multivariate regression we get that the coefficient is positive, a result of the complex correlational structure of the variables mentioned before.

Changes to the text: We have added a new analysis of the univariate models in the Supplementary Note 7 to discuss the correct interpretation of the results of our models for home and mobile food environments and fraction of visits to FFO as a function of demographic variables.

Reviewer Point P 2.6 — Lines 103-104. μ_i captures a heterogeneous measure across individuals and as such, you should avoid reporting it with average and standard deviation. Moreover, this is a quite important measure whose distribution should be plotted and discussed, as it influences the following logistic models and intervention strategies. Ideally, the results should be commented under the lens of extreme value theory.

Reply: The measured fraction of FFO choices to FO options by individual, μ_i , is indeed heterogeneous across users, but not so much. We reported the mean and standard deviation to give a sense of that mild heterogeneity. To fully describe it we have added new information in Supplementary Note 1, including its distribution (see Supplementary Figure S1). Its form is not particularly

strange to human behavior observations, it is bounded and it is not particularly ill-behaved: apart from having a significant number of users that never visit FFOs (22.9%), most of the distribution around small values and with some small fraction of individuals with larger values of μ_i .

In any case, we note that the values of μ_i are not used in our logistic models and intervention strategies. Thus, our models are not influenced by the mild heterogeneity of μ_i . We only use the actual values of μ_i in our model for the DMV visits. Note that we don't approximate μ_i by its mean and standard deviation in that case, but we use their actual values.

In summary, since our main results do not use the actual values of μ_i , we believe our results do not need any other interpretation under the lens of extreme value theory.

Changes to the text: We have added a description of the dataset (including the distribution of μ_i) to the Supplementary Note 1.

Reviewer Point P2.7 — Lines 89-117 and SI Section 7.1. Before drawing conclusions on the R^2 provided by a simple linear model, the authors should consider applying GLM with proper data transformation. Even other machine-learning-based non-linear regressors should be investigated (e.g., random forests).

Reply: The purpose of the models we used in that section was to show that there are no significant differences in the home and mobile environments and the fraction of visits to FFO across demographic groups. We tested that hypothesis using linear models, similar to the Pearson correlation. But the reviewer is correct to point out that the differences between demographic groups could be non-linear. To test that possibility, we have implemented random forest models similar to linear regressions. Since many users share the same ϕ_i^h for the same demographic variables, we have trained the model for ϕ_i^h using a set of home census tract different than the ones used in testing the model to prevent overfitting in that case. Our results for non-linear regressions are very similar to the linear ones: using random forests, we get $R^2 = 0.287$ for ϕ_i^m , $R^2 = 0.057$ for ϕ_i^h , and $R^2 = 0.054$ for μ_i . Although performing slightly better than linear models, our results show that demographic variables have small explanatory power to understand ϕ for both the home and mobility environments. More importantly, more sophisticated models show the small dependence of visits to FFO (μ_i) with demographic variables. This new analysis strengthens our previous conclusions, and we thank the reviewer for raising this important point.

Changes to the text: We have included the random forest model results for home and mobile food environments and the fraction of visits to FFO in the Supplementary Note 7.1 and added a sentence in the main text to reference those results.

Reviewer Point P2.8 — Given the high granularity of the data characterizing food outlets (SI Section 3), the authors should consider stratifying their results and intervention for specific fast-food chains. If this is challenging, the authors should clearly discuss the reason.

Reply: Similar to other works and reports by health agencies like CDC, our objective is to study the overall behavior of users regarding fast-food visits. Although it could be interesting to stratify the results by specific fast-food chains, we could lose the statistical power of our analysis because each particular chain only gather a small fraction of the total FFO in the area. Furthermore, we don't see any bias of users' behavior toward particular brands. In our data, we find that users

visit on average 10.28 ± 5.6 different fast food chains. This result shows that users consume fast food in many different chains and it justifies the holistic perspective of studying the behavior regarding fast-food venues. Finally, another reason not to study a particular chain is that public health initiatives to reduce FFO outlets or increase healthy options would be very unlikely to target specific chains. More successful and already implemented initiatives tend to address this type of retail food business as a whole.

Nevertheless, this idea could be a future avenue for a specific more qualitative study that is beyond the scope of our objective.

Reviewer Point P 2.9 — **What is the degree of universality of the results presented for fast-foods? In other words, if the focus is on different food outlets, what do you expect to observe? Please expand the discussion accordingly.**

Reply: We thank the reviewer for these important questions. Our methodology could be extended to study other food choices, like visits to grocery stores. However, these food behaviors are likely to be different from the ones we study (visits to fast food), some have been studied less in the literature, and they have been fewer policies and funding targeted to them. For example, there is work looking at supermarket access and visits, and policies to improve this, but these are very different food-seeking behaviors with different policy considerations. Access and visits to other types of food outlets, like convenience stores or discount stores, are much less studied.

Given the impact of mobile food environments on visiting fast food outlets, we also expect a similar impact on visiting other types of restaurant/prepared food retail outlets. Thus, following the reviewer's suggestions, we have investigated the effect of mobile food environments on visits to Asian and Latin American food outlets. Results are presented in the new Supplementary Note 11, where we can see that the results are very similar to visits to fast food outlets. These results reinforce the paramount role that mobile food environments may have in diverse food choices. Also, they signal possible holistic policies to improve population diet quality. This could be an avenue for future research but is beyond the scope of our present study.

Changes in the text: We have included a new note in the Supplementary Material to analyze the degree of universality of our results to other food choices and expanded the presentation of the results and the discussion accordingly.

Reviewer Point P 2.10 — **Line 223. Converting a FFO into a non-FFO could work theoretically but it is worth exploring how this simple intervention translates to reality. According to what you have learned about human behavior and mobility, are you able to come up with more practical suggestions? Is it possible to identify which non-FFO is ideal for a given location?**

Reply: We thank the reviewer for the thoughtful comment. We agree that the practicality of the specific interventions we investigate can be a challenge. However, we want to clarify that the focus of the simulation study is not to identify exactly 'how we can convert this specific FFO to a healthy outlet', and what type of 'healthy' outlet that might be. Rather, we focus on the theoretical exercise of identifying optimal locations in which to *change the overall ratio between FFO and healthier options* in an area by one unit. We see this theoretical exercise as being able to inform or inspire the creation of specific policy levers or incentives for impacting the overall ratio of FFO to non-FFO (or to all FFO) in an area, which should be investigated in a future policy-specific analysis.

Although in our paper we use the analogy of changing one FFO to non-FFO in an area, our intervention is, in fact, related to a change of the FFO to FO by one unit. We now discuss several strategies for accomplishing that, each with examples of implementation in practice: decreasing the number of FFO relative to non-FFO; increasing the number of non-FFO relative to FFO; and specifically converting FFO to non-FFO. Existing policy strategies are more likely to involve one of the first two approaches, as outlined below.

- Regarding decreasing the number of FFO relative to non-FFO, this kind of intervention has been successfully implemented by several community governments through the introduction of policies banning the opening of new FFO [1, 13, 14]. In a study by Brown et al. (2022) [1], the authors document how a banning new FFO in a region can lead to a sustained change in the proportion of FFO with respect to all food outlets, ϕ . In their specific case study in the UK, a blanket ban in a district led to a reduction of 15% in ϕ within 2 years of the ban implementation going from 45% to 30%. Nixon et al. (2015) [13] document the proposal and responses to policies implementing the ban of new FFO in the US. They calculate that close to 30% of the proposed bans in their sample were rejected.
- Regarding increasing the number of non-FFO relative to FFO, this could be operationalized through policies investing resources to support non-FFO to obtain business licenses or opportunities in food retail. Winne et al. (2005) [18] suggest several such options. An example in practice is the US federal government's subsidization of the new food chain start-up Everytable [8], which aims to sell "nutritious, fresh, made-from-scratch food, at fast-food prices" at its franchises in a similar business model as a fast food chain. In an investment policy being promoted as one of the White House's nutrition strategies, Everytable is receiving investment from White House nutrition initiatives to support local entrepreneurs to become Everytable franchise owners [15].
- Converting a FFO to a non-FFO might be seen as the least practical intervention to implement since interfering with businesses in operation goes beyond the scope of most food policy interventions. There is one exception that we are aware of: when an existing FFO changes its menu to provide healthier options, thereby shifting its balance of 'fast food' items (i.e., energy-dense, low nutrient, highly processed foods) to healthier 'non-fast food' items (e.g., items including whole grains, vegetables, and fruits). A precedent for a FFO chain introducing healthier menu items in response to government policy has been set by restaurants seeking to participate in the "Restaurant Meals Program" (RMP), which is part of the national Supplemental Nutrition Assistance Program (SNAP). In general, FFO are not eligible to participate in SNAP since the program is intended to help low-income individuals and families purchase groceries for home consumption. However, the RMP, operating in certain US states, is an exception to this rule, allowing qualifying restaurants to accept SNAP benefits from eligible customers. While specific requirements for qualifying for the RMP vary by state or county, in general, certain nutritional criteria must be met; these include offering healthy menu options that meet specific requirements for limits on calories, fat, and sodium and minimum standards for protein, vitamins, and other nutrients [9]. In Los Angeles County, for example, restaurants qualifying for the RMP must have at least five healthy meal options that meet a specific nutritional value [5]. These requirements have incentivized fast food chains to adapt their menus to include several healthy items. While this is not the level

of change that would be required to shift the balance of a menu all the way from 'FFO' to 'non-FFO', it represents a strategy in that direction.

Regarding the body text, we agree with the reviewer that the text does not appropriately introduce and clarify the challenges described above. We have therefore added to the text introducing the simulation study (copied below for the reviewer's convenience). In the adapted text, we clarify the point that the simulation exercise is theoretical in nature while providing several examples of policies that have been implemented in practice to shift the balance of FFO to non-FFO in an area in order to improve the food environment. Again, because the exercise is theoretical, we do not go into details about these intervention strategies and how they might best impact the dietary health of those using these food environments, including what the non-FFO should be. We, therefore, suggest in the Discussion that future policy-focused work investigate these aspects of the implementation of the strategies we experiment with in this work.

Changes to text: We have modified the definition of our intervention from "changing one FFO to non-FFO" to "increase the ratio of non-FFO to FFO by one unit" to reflect better the more general character of our intervention and potential practical suggestions to accomplish this. We have added the following paragraphs to the *Policy Implications* section:

Changing the ratio of FFO to FO in an area might be accomplished by various strategies or interventions: including decreasing the number of FFO relative to non-FFO, increasing the number of non-FFO relative to FFO, or converting a FFO into a non-FFO. Policy strategies in practice are more likely to involve one of the first two approaches, such as implemented interventions that have banned new FFO from opening in a specific area [1, 13, 14], or supported non-FFO to obtain business licenses or opportunities in food retail [18]. The former was implemented in some countries (UK), where a blanket ban in a district led to a reduction of 15% in $\phi(x)$. The latter was recently exemplified by a US White House nutrition initiative that invested in programs supporting local entrepreneurs to open healthy prepared-food outlets [15].

To illustrate this, we consider a simple intervention in which we increase the ratio of non-FFO to FFO by one unit in a particular area. In this case, $\phi(c_{it})$ changes by approximately $\delta\phi/\delta I \simeq -1/n_{\Omega}$ where n_{Ω} is the number of FO in the area. We see this strategy as a theoretical illustration of the general concept of improving the food environment in an area by shifting the balance of FFO to healthier outlets by the same amount of effort per area. If the methodology we develop in the following were to be used, this could be implemented by setting the derivative of ϕ in our formula to be the expected derivative of the intervention. We have also extended the model to all times of the day and week to describe the full effect of the strategy (see Methods). Assuming that we have limited resources to change hundred food outlets, where are the areas in which our intervention maximizes its impact? Here, we compare four different strategies.

and the following one to the *Discussion*:

Finally, future policy-specific work should seek to investigate further details of the operationalization and implementation of the theoretical strategies we investigate here for

shifting the balance of FFO to non-FFO in a food environment to optimize the dietary health of individuals frequenting that environment, including specifically how and what kinds of outlets should be involved in the interventions.

Reviewer 3

Reviewer Point P 3.1 — The manuscript seeks to address an important research need in food environment research, addressing restaurants beyond commonly used static models. The use of mobile data is innovative. Yet, due to this, the explanation of the methods and approach, while comprehensive, missed some important details to evaluate the information presented.

Information about the participants was lacking or not clear. While the data were anonymous, it would have helped to know the overall demographics and key aspects of the people providing the data. Some aspects include age, gender, and employment status, at minimum, as factors that would influence the information collected.

Reply: This is an important point. For privacy reasons, we don't have demographic information about the individual users. We can only proxy their demographic characteristics by using the ACS census data by using the demographic characteristics of the population living in the same area. In Supplementary Note 2 we have a comparison of how representative is our sample of users with respect to the population living in the urban areas. A more extensive comparison of the demographic representativity of our sample of users was done in reference [12] (reference [17] in the main paper), where we saw that our sample was a slightly biased towards high-income people. Note however, that our results are largely independent of that bias: in the Supplementary Note 2 we have used post-stratification methods to alleviate potential demographic biases and we found that our main results (Figures 3 and 4 in the main paper) are very similar to the original ones in the main paper. In this sense, we are confident that our results are representative of the population living in the metropolitan areas studied.

Reviewer Point P 3.2 — Some aspects of the analysis were not well justified. For example the DMV experiment visit. In practice, I wonder about the choice, given that people may eat before or after, not really having to eat in the area (as opposed to a place of employment).

Reply: The DMV analysis was included to complement our previous analysis, specifically to show that the effect of the context is also important even for contexts that are not chosen for their particular food environment. Specifically, we expect no relationship between a person's food preference, and the food environment the person is exposed to in the DMV. Note that the DMV analysis is exactly the same analysis we did around lunch, but conditioning on only those food actions where the pre-lunch context was the DMV. In fact, the DMV data is included in the first analysis (together with the rest of the pre-lunch contexts). Thus, we are only studying those decisions to go or not to a FFO, conditioning on going for lunch just after visiting the DMV. We have modified the main text to explain that the analysis around DMVs is the same as the original analysis but restricting to contexts around DMV.

Changes to text: We have modified the main text to stress that the analysis around DMV is just a representative case of the general analysis to investigate the effect of food environments even when contexts are not related to a person's food preferences.

Reviewer Point P 3.3 — Some aspects of the policy simulation needed additional clarification. For example, the discussion of changing FFO to non-FFO: what are these non-FFO? Are these restaurants? Or other establishments?

Reply: We thank the reviewer for this comment. Following this comment and the response to point P 2.10 we have revised the definition of our policy in the main text. Specifically, “we consider a simple intervention in which we decrease the ratio of FFO to FO by one unit in a particular area”. This reflects better the more general character of our intervention and potential practical suggestions to accomplish this. Point P 2.10 shows different routes to accomplish this simple intervention, including changing a FFO to a non-FFO which are also food outlets.

Reviewer 4

Reviewer Point P4.1 — First, the study is far from being reproducible, the data are only available upon request submitted to Cuebiq. I followed the link provided by the authors, and, after several attempts, it seems that it will be quite complicated and time consuming to get similar Cuebiq/spectus data to that used by the authors to perform the analysis. The other data used (American Community Survey and PLACES Local Data) should also be clearly identified. Finally, the link provided by the authors to reach the code used to run the analysis is not available and there is no `mobile_food_environments` repository on emoro GitHub account. . . In my opinion, the paper does not meet the current data (and code) sharing standard for publication. I can therefore not recommend this paper for publication without a permanent link to access the data used to perform the analysis.

Reply: Our previous Data Statement followed closely Nature Editorial Policies (see <https://www.nature.com/nature-portfolio/editorial-policies/reporting-standards>), regarding the use of data with controlled access. Data is available from Spectus (formerly known as Cuebiq) through their Social Impact program upon request (<https://spectus.ai/lp/book-a-demo/>). As requested in the submission process, we disclosed that information about controlled access in our original submission.

Nature Group journals (including Nature Communications) have published many papers from many other groups on the same dataset (and many others datasets with controlled access from private companies) declaring similar data availability restrictions and statements (see <https://doi.org/10.1038/s41467-023-36985-0>, <https://doi.org/10.1038/s41467-022-35052-4>, or <https://doi.org/10.1038/s41467-022-34592-z> for a sample of papers with similar data statements in the last 3 months).

Nevertheless, during the last months, we have worked together with our data provider (Cuebiq, now Spectus) to make the data used for our analysis in the manuscript accessible to readers and researchers in a more direct way. The data used in our analysis is now available on request from <https://doi.org/10.5281/zenodo.7798632>.

For the other data used, the manuscript already had references to the web pages where the data is publicly available. Specifically, the American Community Survey can be downloaded from <https://www.census.gov/programs-surveys/acs> (it was identified in our former reference [65]), the PLACES Local Data can be accessed at <https://www.cdc.gov/places/about/500-cities-2016-2019/index.html> (it was already identified in our former reference [66]), and the Food Desert dataset can be downloaded from the U.S. Department of Agriculture's Economic Research Service at <https://www.ers.usda.gov/data-products/food-environment-atlas/> (it was already identified in our former reference [8]). To facilitate access and identification to those datasets, we have placed those links directly in the *Data availability* paragraph. We have also added more information about those datasets in Supplementary Note 1.

We apologize for not making the repository with our code public by the time of the review process. We have now made it public and can be found in the same direction https://github.com/emoro/mobile_food_environments

Changes to the text: We have modified the Data Statements and Code Availability Statements to reflect the newly available data. We have also included more information about the mobility data and other datasets used in the Supplementary Note 1.

Reviewer Point P 4.2 — Second, I would recommend the authors to give more details about the data and the cleaning process (algorithms and parameters used to identify the home locations, FO/FFO visits...). I maybe missed something but it is not clear to me how the authors have identified the FO/FFO visits. Is it based on a check-in or individual presence in the building? Most of the findings are based on these “visits”, I would encourage the authors to give more details regarding the methodology used to identify these visits and to perform a thorough analysis of FO and FFO daily visits at an individual level (number of FO or FFO visits per user, day and census track, identification and validation of individual routines...).

Reply: As mentioned in Supplementary Materials, the data, algorithms, and methods to obtain the visits are the same as we used in our previous works [11, 12]. Our methods are very similar to the ones typically used in the literature on semantic analysis of trajectory data [10]. Both our previous works contain extensive material about the robustness of the methods used, the representativity of the data, and the comparison and validation of our data with other datasets. In particular, as found in [12], our sample of users and their behavior is highly representative of the people living in those metropolitan areas, including their socio-demographic profile and the visitation patterns to different amenities. Also, in [11], we found that neighborhood-level features representing visits to fast food outlets (FFO) were significantly associated with self-reported fast-food intake, significantly associated with obesity and diabetes, and were a better predictor of these diseases than self-reported fast food intake.

Although the Supplementary Material contained already extensive information on how visits and homes are determined, to make this even clearer for the reader, we have expanded the Supplementary Material of our manuscript to incorporate more information in the Supplementary Note 1, specifically about how stays are detected, how they are attributed to different POIs and how homes are determined.

Regarding FO/FFO visits, the Supplementary Material contained extensive information about how food outlets are classified (Supplementary Note 3) and how robust our results are to that classification (Supplementary Note 12.4). We also had information about how a visit to a FO and FFO is determined including tests for examining how robust our results are to that method (Supplementary Note 12.1). Nevertheless, we have expanded Supplementary Note 1 to include more information about how FO/FFO visits are determined.

Finally, we have added a more descriptive analysis of the data in the Supplementary material to help the reader understand better the heterogeneity between individuals and across stays.

Changes to the text: We have added new sections to describe the details about the data and the cleaning processes and algorithms to detect stays, identify home location, FO/FFO visits in Supplementary Note 1 and references to them in the main text. We have also included a new descriptive analysis of the data in the Supplementary Material.

Reviewer Point P 4.3 — Finally, I would recommend the authors to add a control mechanism to handle potential problems of spatial autocorrelation. Indeed, spatial context and FO/FFO visits can be spatially linked if the distance between context and visit is too small (as shown in Table S3).

It would recommend the authors to used an increasing d_{min} instead of a d_{max} to clearly assess the effect of spatial correlation.

Reply: In our analysis, the context before lunch and the place where lunch happens do not have to coincide. In our data, a large fraction ($\sim 60\%$) of food outings happen very close to the contexts ($d_{it} < 3km$, where d_{it} is the distance between c_{it} and the place chosen for lunch), so we might expect a direct influence of the context on the decision made, which is precisely what we want to measure: what is the effect of the food environment users are exposed when making the decision to visit FO/FFO.

But the reviewer might suggest that the spatial correlation of the ratio of FFO to FO might extend that influence beyond the neighborhood of the context. This assumption is based on the idea that FO/FFO contexts are spatially correlated. However, our data shows that this is not the case. We observe that the spatial correlation of $\phi(x)$ decays dramatically after 2km (see Figure S6). Thus, our results are not affected by potential correlations of the spatial contexts.

Nevertheless, we have changed our Supplementary Note 7.2 to discuss in a more general way the role of distance in our model, as the reviewer suggests. Instead of running a model for each distance threshold, we investigate the effect of the distance by including it as a variable in our regression. This way, we have more statistical power, and the model better reflects the whole behavior of people that can be different close to the context or beyond it. Our results show that, indeed, distance is modulating the effect of food environments: when the food choice is made far away from the pre-lunch context, the food environment has a smaller effect on people's decisions. However, for the majority of our data ($d_{it} < 3km.$), we still get that the impact of the context is positive and significant, as in the result in the main paper.

Changes to the text: We have revised our Supplementary Note 7.2 to discuss the potential problem of the spatial autocorrelation of ϕ and the impact of the distance on our model. We have also commented on the effect of the distance on our results in the text and Methods of the main paper.

Other changes to the text

Apart from the changes in the text to address the comments of the reviewers, the new version of the manuscripts contains some small changes:

- While preparing the data and code to share, we discovered a coding mistake in the preparation of Figure 1b and Figure 2b that discarded a small random fraction of visits to FO (below 2%). This small fraction of visits is now included in the analysis of Figures 1 and 2 and in the medians of distances to different types of venues in the text. None of our conclusions and analysis are affected by this rectification, and the changes in both figures are inappreciable.
- Finally, when computing the median distance to grocery stores in kilometers, we incorrectly divided the distance in meters by one order of magnitude more. It is not corrected in the Results section. Instead of being $0.4km$ it really was $4km$. Note that the numbers are now different after the change mentioned in the first point above.
- Also, to have more statistical power (especially for Figure 3b), we increased the lunchtime hours from 11h30 to 14 local time. This change does not alter our main results in the body of the paper and Supplementary Material. But, of course, the numerical values in Figure 3 and Figure 4 are different.
- We also discovered a small mistake in the visualization code in Figure 4a regarding the low-income, low-access areas, which is now corrected. Our main results in the text and the Supplementary Material regarding interventions are not affected by that visualization mistake.
- We have also thoroughly revised the paper to remove some typos and grammar mistakes.

We want to reiterate that these minor corrections have no impact on the results or interpretation of our paper.

References

- [1] Heather Brown, Huasheng Xiang, Viviana Albani, Louis Goffe, Nasima Akhter, Amelia Lake, Stewart Sorrell, Emma Gibson, and John Wildman. No new fast-food outlets allowed! evaluating the effect of planning policy on the local food environment in the north east of England. *Social Science & Medicine*, 306:115126, 2022.
- [2] Thomas Burgoine and Pablo Monsivais. Characterising food environment exposure at home, at work, and along commuting journeys using data on adults in the UK. *The International Journal of Behavioral Nutrition and Physical Activity*, 10(1):85–85, 2013.
- [3] Andreea Cetateanu and Andy Jones. How can GPS technology help us better understand exposure to the food environment? A systematic review. *SSM - Population Health*, 2:196–205, 12 2016.
- [4] Basile Chaix. Mobile sensing in environmental health and neighborhood research. *Annual review of public health*, 39:367–384, 2018.
- [5] Evan Cohen. *The Restaurant Meals Program: A Guide for New York State's Successful Implementation*. PhD thesis, Rochester, New York, 2022.

- [6] Tamara Dubowitz, Shannon N Zenk, Bonnie Ghosh-Dastidar, Deborah A Cohen, Robin Beckman, Gerald Hunter, Elizabeth D Steiner, and Rebecca L Collins. Healthy food access for urban food desert residents: examination of the food environment, food purchasing practices, diet and bmi. *Public health nutrition*, 18(12):2220–2230, 2015.
- [7] Jeanette Eckert and Igor Vojnovic. Fast food landscapes: Exploring restaurant choice and travel behavior for residents living in lower eastside detroit neighborhoods. *Applied Geography*, 89:41–51, 2017.
- [8] Everytable. Everytable. <https://www.everytable.com/>, 2023. Accessed: 03-20-2023.
- [9] USDA Food and Nutrition Service. Supplemental Nutrition Assistance Program Restaurant Meals Program Information and Eligibility Requirements. <https://www.fns.usda.gov/snap/retailer/restaurant-meals-program>, 2023. Accessed: 03-20-2023.
- [10] Ramaswamy Hariharan and Kentaro Toyama. Project lachesis: parsing and modeling location histories. In *International Conference on Geographic Information Science*, pages 106–124. Springer, 2004.
- [11] Abigail L. Horn, Brooke M. Bell, Bernardo Garcia Bulle Bueno, Mohsen Bahrami, Burcin Bozkaya, Yan Cui, John P. Wilson, Alex Pentland, Esteban Moro, and Kayla de la Haye. Investigating mobility-based fast food outlet visits as indicators of dietary intake and diet-related disease. *medRxiv*, 2021.
- [12] Esteban Moro, Dan Calacci, Xiaowen Dong, and Alex Pentland. Mobility patterns are associated with experienced income segregation in large US cities. *Nature Communications*, 12(1):4633, 2021.
- [13] Laura Nixon, Pamela Mejia, Lori Dorfman, Andrew Cheyne, Sandra Young, Lissy C Friedman, Mark A Gottlieb, and Heather Wooten. Fast-food fights: news coverage of local efforts to improve food environments through land-use regulations, 2000–2013. *American journal of public health*, 105(3):490–496, 2015.
- [14] Roland Sturm and Aiko Hattori. Diet and obesity in Los Angeles County 2007–2012: Is there a measurable effect of the 2008 “Fast-Food Ban”? *Social Science & Medicine*, 133:205–211, 2015.
- [15] The White House Briefing Room. Fact sheet: The Biden-Harris administration announces more than 8 billion in new commitments as part of call to action for White House conference on hunger, nutrition, and health. <https://www.whitehouse.gov/briefing-room/statements-releases/2022/09/28/fact-sheet-the-biden-harris-administration-announces-more-than-8-billion-in-new-commitments-as-part-of-call-to-action-for-white-house-conference-on-hunger-nutrition-and-health>, 2022. Accessed: 2023-03-20.
- [16] United States Census Bureau. 2013-2017 American Community Survey 5-year Estimates. <https://www.census.gov/programs-surveys/acs>, 2019. Accessed: 2020-12-04.
- [17] Michele Ver Ploeg, Lisa Mancino, Jessica E Todd, Dawn Marie Clay, and Benjamin Scharadin. Where do Americans usually shop for food and how do they travel to get there? initial findings from the national household food acquisition and purchase survey. Technical report, 2015.
- [18] Mark Winne. Community food security: Promoting food security and building healthy food systems. *Venice, CA: Community Food Security Coalition*, 2005.

REVIEWER COMMENTS

Reviewer #1 (Remarks to the Author):

I was happy to read the very detailed response by the authors. The most crucial points I raised were due to lack of clarity in the exposition. I can see that has been fixed, even though the new version of the manuscript is still a rather heavy read.

I am overall satisfied with the changes and the couple of additional neat sanity checks and experiments.

Regarding point P1.7, which was the most critical one I wanted to clarify, I get the explanation in the rebuttal, and it makes sense. The additional test that pointed out the different attitude to eating out when at home or on the move is also an interesting additional signal that I am glad the authors found.

I think that the paper is ready for publication.

Reviewer #2 (Remarks to the Author):

I would like to thank the Authors for addressing my previous points and revising the manuscript. Unfortunately, it has been quite challenging to spot all the changes with no highlighting of the text (and I'm aware you refer to the text changes in your responses).

A few points to address:

- P 2.2: Thank you for investigating the results for different cities. These are the kind of data stories I initially mentioned in my previous review. I think discussing your outcomes in a more extensive fashion in the manuscript (not just at line 277 and in the SI) would make your paper relevant to a broader audience and would clearly show the strength of your approach. For example, Figure S14 is quite interesting and while the trend observed is expected, differences between cities with \sim the same (for example Boston vs Philadelphia, correct me if I got the color-coding wrong) are worth discussing in the main. It gives valuable practical insights.

- P 2.4: Please clearly state the number of data points represented in the density plots and calculate and discuss the differences in effect sizes in comparing different distributions (e.g., with Mann Whitney U-Rank test, see effect size calculations for behavioral sciences)

- Line 91: Stratify the positive correlation between ϕ_m and ϕ_h according to demographics and other covariates and discuss the results per group.

- Line 104-105 "This suggests that socio-demographic inequalities propagate stronger to people's experienced mobile food environments than to their food spatial accessibility at home": this is quite interesting and deserves more discussion. The authors are experts in mobility: does mobility exacerbate inequalities beyond food? Is this a universal feature?

- P 2.6: Thank you for the clarification (I was referring to the DMV model). Nevertheless, please avoid expressing a skewed distribution of non-negative numbers with average and standard deviation, especially if the range including a symmetric standard deviation becomes negative. Use median and IQR at Line 110.

- It is interesting how at Line 132 0.268 ± 0.001 is discussed as a strong positive correlation, while at Line 91 0.213 ± 0.001 is reported as small. Could you please elaborate and maybe rephrase?

-

Overall, I found the paper improved and I appreciate the effort in providing detailed responses.

Reviewer #4 (Remarks to the Author):

First of all I would like to thank the authors for these additional efforts, especially with regard to the data and code sharability. Beyond journal policies, reproducibility in science is a very important topic and being able to reproduce the analysis and figures of a paper based on the raw data and the code provided by the authors will hopefully be a standard in a few years. I understand that this is a period of transition toward fully reproducible science so I really appreciate the effort made by the authors to share their data by providing a (restricted access) zenodo link and a github repository hosting the codes used to produce the figures from these data.

Regarding my second concern, the method used to extract the stays and to assign them to a POI is perfectly clear now. However, I maybe missed something but it is not clear to me how did the authors assess the reliability of the assignment? I understand that it is complicated to evaluate if a user staying near a food outlet during lunch time was actually eating in this food outlet but I would encourage the authors to discuss this point further in a dedicated methodological discussion in the main text (by adding a section "Limitations of the study" in the discussion for example).

Finally, I would like to thank the authors for the additional Figure S6. If I got it right, Figure S6 shows the correlation of FFO/FO ratio between contexts (circle of 1km radius?) as a function of the (euclidean?) distance between contexts. The correlation is very high for small distances (lower than 1km) presumably due to an overlap between contexts (?). Therefore, the distance between context and action plays an important role in the analysis. I would encourage the authors to add a table and a discussion in the main text regarding the impact of the distance on the finding "when the context includes 10% more FFO, there is an increase in the odds to visit an FFO of 20%". The table could include different class of distances (lower than 1km, between 1 and 2 km, between 2 and 3 km and more than 3km) and the odds ratio associated to each class. Another way of controlling the impact of spatial autocorrelation would be to apply the model and compute the odds ratio for different class of FFO/FO ratio at destination (i.e. FO visit).

Response to the reviewers

Once again, we thank the reviewers for their supportive, critical, and extensive assessment of our work “You are where you eat: Effect of mobile food environments on fast food visits,” submitted for consideration for publication in Nature Communications (reference #NCOMMS-22-32906A-Z).

In the following, we address their new concerns point by point. We also specify the changes made to the main paper and the Supplementary Material to address them.

Reviewer 1

Reviewer Point P 1.1 — I was happy to read the very detailed response by the authors. The most crucial points I raised were due to lack of clarity in the exposition. I can see that has been fixed, even though the new version of the manuscript is still a rather heavy read.

I am overall satisfied with the changes and the couple of additional neat sanity checks and experiments.

Reply: We greatly appreciate your constructive and positive comments.

Reviewer Point P 1.2 — Regarding point P1.7, which was the most critical one I wanted to clarify, I get the explanation in the rebuttal, and it makes sense. The additional test that pointed out the different attitude to eating out when at home or on the move is also an interesting additional signal that I am glad the authors found.

Reply: We thank the reviewer for her/his comment (1.7), which allowed us to find this important and additional signal in attitudes to fast food visits around home and beyond. In turn, this made our results stronger and more nuanced.

Reviewer Point P 1.3 — I think that the paper is ready for publication.

Reply: Again, we really thankful for your constructive and positive review of the previous and current versions of our paper.

Reviewer 2

Reviewer Point P 2.1 — I would like to thank the Authors for addressing my previous points and revising the manuscript. Unfortunately, it has been quite challenging to spot all the changes with no highlighting of the text (and I’m aware you refer to the text changes in your responses).

Reply: We thank the reviewer for your constructive and positive comments. In this revision, we have highlighted the changes to the text to make it easier to spot all changes.

Reviewer Point P 2.2 — A few points to address: - P 2.2: Thank you for investigating the results for different cities. These are the kind of data stories I initially mentioned in my previous review.

I think discussing your outcomes in a more extensive fashion in the manuscript (not just at line 277 and in the SI) would make your paper relevant to a broader audience and would clearly show the strength of your approach. For example, Figure S14 is quite interesting and while the trend observed is expected, differences between cities with ~ the same (for example Boston vs Philadelphia, correct me if I got the color-coding wrong) are worth discussing in the main. It gives valuable practical insights.

Reply: As we mentioned in our previous revision, our objective in the paper is to find the commonalities of FFO visits in major urban areas and the impact of mobile food environments. And, indeed, we find a very strong common behavior of individuals across the 11 cities analyzed. We believe the strength and commonality of our results already make our paper relevant to a broader audience since it shows that the impact of mobile food environments and in turn, behavioral interventions, generally apply to the whole population living in the major urban areas in the US (83 million people in our study).

We agree with the reviewer in that the analysis of tiny deviations from that common behavior could be very important (e.g. the 15% difference between the average effect in Philadelphia and Boston in Figure S14) for specific interventions in small areas, but it is beyond the scope of this paper. We plan to pursue this line of research in future investigations.

We believe those small differences are probably due to city-specific demographic, structural or commercial characteristics. As we show in the intervention section, behavioral interventions are more effective in areas related to "Malls", "Industry / Factory", etc., so the small differences observed could be due to cities having a different composition of those areas.

Changes to the text: We have added the following sentence when describing the results by city at the end of :

"Behavioral interventions in cities with different relative compositions of those areas might have slightly separated effects. For example, although behavioral interventions are always more effective than static interventions, the difference is slightly smaller in cities like Philadelphia, Seattle, or Los Angeles, suggesting that behavioral interventions depend somewhat on city-specific structural or commercial characteristics."

Reviewer Point P 2.3 — - P 2.4: Please clearly state the number of data points represented in the density plots and calculate and discuss the differences in effect sizes in comparing different distributions (e.g., with Mann Whitney U-Rank test, see effect size calculations for behavioral sciences)

Reply: Again, we note that the difference between the distributions in Figure 1B is not used anywhere in the paper. We only show those distributions to see that most of the visits to food and fast food outlets happen very far away from individuals' home census tracts.

Moreover, the number of data points used in Figure 1b was clearly stated in the Data Section in the Supplementary Material (see Table S1). They vary from 245.05 million points for all visits and 62.04 million points for food visits, to 10.81 million points for fast food visits. We have included in the Supplementary Material the result of the Mann-Whitney U-Rank test and the effect size for the comparison between all visits and visits to food places. As expected, all differences are statistically significant and the effect size between all places and grocery stores is the largest. We have added a reference to these analyses in the main text too.

Reviewer Point P 2.4 — - Line 91: Stratify the positive correlation between ϕ_m and ϕ_h according to demographics and other covariates and discuss the results per group.

Reply: Thank you for this new comment. We have included an analysis and new material about the stratification of the positive correlation between ϕ_m and ϕ_h . Specifically, we found “that correlation is slightly different across demographic groups, with residents in areas with low-income, high percentage of Black population, or large use of public transportation having a little bit more correlation between home and mobile environments. But in general, the correlation is small across groups $\rho \leq 0.29$ (see Supplementary Figure S19)”. Thus, because of those small differences, our main observation in this section, that “the food environments that users are exposed to throughout the day are different from the ones around their homes” is still valid.

Since the main results in our paper are done individually, these slight differences are already accounted for in the model in Equation (1) and, more importantly, in the interventions section. Thus, our results do not depend on these slight demographic differences.

Changes to the text: We have added the following sentence to the discussion after line 91: “that correlation is slightly different across demographic groups, with residents in areas with low-income, high percentage of Black population, or large use of public transportation having a slightly stronger correlation between home and mobile environments. But in general, the correlation is small across groups $\rho \leq 0.29$ (see Supplementary Figure S19)”

Reviewer Point P 2.5 — - Line 104-105 “This suggests that socio-demographic inequalities propagate stronger to people’s experienced mobile food environments than to their food spatial accessibility at home”: this is quite interesting and deserves more discussion. The authors are experts in mobility: does mobility exacerbate inequalities beyond food? Is this a universal feature?

Reply: Thank you for this new comment. What we found is that mobile food environments users are exposed to can be described slightly better than home food environments using socio-demographic characteristics of individuals. When people move around in the city they have more choices and opportunities to visit different places. Also, most of the food visits happen far away from home. We also need to bear in mind that these preferences/opportunities are shaped by differential access to healthy foods in the first place, targeted marketing, and other social and structural forces. Thus, we expect that their preferences and opportunities (intertwined with their demographic characteristics) are better expressed in their actions than in where they live.

This behavior is also expected in other aspects of urban life. For example, visits to cultural venues are better described by demographic variables of people visiting those places, rather than people that live around those areas. In general, it is well-accepted that mobility is different across gender, income, race and many other demographic traits. We agree with the reviewer that the general study or potential universality of how mobility allows individuals to express better individual preferences or opportunities in different aspects of their life is very interesting, but it is beyond the scope of this paper.

In any case, demographic characteristics only explain a small fraction of the characteristics of mobile or home food environments. We have modified the main text around this to acknowledge. We have also changed the word “inequalities” to “differences” since the former typically implies other social processes far away from what we want to describe in that section.

Changes to text: The sentence mentioned by the reviewer has been changed to "This suggests that socio-demographic differences propagate slightly more strongly to people's experienced mobile food environments than to their food spatial accessibility at home, probably because those demographic traits shape slightly the differential access to different environments, targeted marketing, and other social and structural forces."

Reviewer Point P 2.6 — - P 2.6: Thank you for the clarification (I was referring to the DMV model). Nevertheless, please avoid expressing a skewed distribution of non-negative numbers with average and standard deviation, especially if the range including a symmetric standard deviation becomes negative. Use median and IQR at Line 110.

Reply: We have changed the value reported to the median and IQR at Line 110.

Reviewer Point P 2.7 — - It is interesting how at Line 132 0.268 ± 0.001 is discussed as a strong positive correlation, while at Line 91 0.213 ± 0.001 is reported as small. Could you please elaborate and maybe rephrase?

Reply: We have rephrased the text around line 132. Our intention was to explain that the correlation $\rho(\phi_i^m, \mu_i) = 0.268 \pm 0.001$ for mobile food environments is stronger compared to the $\rho(\phi_i^h, \mu_i) = 0.068$ for home food environments.

Reviewer Point P 2.8 — - Overall, I found the paper improved and I appreciate the effort in providing detailed responses.

Reply: We greatly appreciate your constructive and positive comments.

Reviewer 3

Reviewer 4

Reviewer Point P 4.1 — First of all I would like to thank the authors for these additional efforts, especially with regard to the data and code sharability. Beyond journal policies, reproducibility in science is a very important topic and being able to reproduce the analysis and figures of a paper based on the raw data and the code provided by the authors will hopefully be a standard in a few years. I understand that this is a period of transition toward fully reproducible science so I really appreciate the effort made by the authors to share their data by providing a (restricted access) zenodo link and a github repository hosting the codes used to produce the figures from these data.

Reply: We thank the reviewer for this point. We agree that providing the data and code to reproduce our results makes our research project stronger. We are glad that we could find a way with our data provider to share the data.

Reviewer Point P 4.2 — Regarding my second concern, the method used to extract the stays and assign them to a POI is perfectly clear now. However, I maybe missed something but it is not clear to me how did the authors assess the reliability of the assignment? I understand that it is complicated to evaluate if a user staying near a food outlet during lunchtime was actually eating in this food outlet but I would encourage the authors to discuss this point further in a dedicated methodological discussion in the main text (by adding a section “Limitations of the study” in the discussion for example).

Reply: We thank the reviewer for this point. We have a number of robustness checks in the Supplementary Material to test how our results depend or not on the visit assignment algorithm. As we discussed in our previous round of responses, we found that our assignment algorithm assumptions do not change the strength and quality of our results.

Furthermore, our manuscript contains a large “Limitations” paragraph at the end of the “Discussion” section in which we comment specifically on the point raised by the reviewer regarding diverse nutritional quality across FFO, problems regarding the multi-story or multi-purpose buildings, drive-throughs, etc. As we say in that section, “Our results, therefore, serve a proxy and are a bound for the potential FF intake”.

Reviewer Point P 4.3 — Finally, I would like to thank the authors for the additional Figure S6. If I got it right, Figure S6 shows the correlation of FFO/FO ratio between contexts (circle of 1km radius?) as a function of the (euclidean?) distance between contexts. The correlation is very high for small distances (lower than 1km) presumably due to an overlap between contexts (?). Therefore, the distance between context and action plays an important role in the analysis. I would encourage the authors to add a table and a discussion in the main text regarding the impact of the distance on the finding “when the context includes 10% more FFO, there is an increase in the odds to visit an FFO of 20%”. The table could include different class of distances (lower than 1km, between 1 and 2 km, between 2 and 3 km and more than 3km) and the odds ratio associated to each class. Another way of controlling the impact of spatial autocorrelation would be to apply the model and compute the odds ratio for different class of FFO/FO ratio at destination (i.e. FO visit).

Reply: We thank the reviewer for raising this important point again. As we discussed in our previous answer to this point, instead of running a model for each distance threshold as suggested again by the reviewer, we investigated the effect of the distance “by including it as a variable in our regression. This way, we have more statistical power, and the model better reflects the whole behavior of people that can be different close to the context or beyond it”. The results of this regression were in Table S6. As we mentioned in our previous answer, “Our results show that, indeed, distance is modulating the effect of food environments: when the food choice is made far away from the pre-lunch context, the food environment has a smaller effect on people’s decisions. However, for the majority of our data ($d_{it} < 3\text{km.}$), we still get that the impact of the context is positive and significant, as in the result in the main paper.”

We have modified the text around the finding “when the context includes 10% more FFO, there is an increase in the odds to visit an FFO of 20%” to account for the modulating effect of the distance to the pre-lunch context as suggested by the reviewer.

Changes to the text: We have included this sentence in the main text “Finally, as expected,

we find that distance between context and lunch choice modulates this effect, although, for the majority of our data, we still get that the impact of context is positive and significant.”

REVIEWER COMMENTS

Reviewer #2 (Remarks to the Author):

No further comments.

Reviewer #4 (Remarks to the Author):

First, I would like to apologize to the authors, I was focusing too much on the Supplementary Information during the last round of review and missed some information already present in the revised version of the main paper.

Regarding my first concern, there is no discussion about the link between a "stays" and an eventual purchase in a POI. This is not because there is a stop in your trajectory near a place that you can guarantee that an activity has been achieved in this place. This is a general issue and there is not much that you can do about it here but I would be grateful if you could add a sentence about that in the "Limitations" paragraph.

I will be less accommodating regarding my second concern. I still believe that the distance between context and lunch choice is really important here and I am not satisfied by the authors' answer. The interpretation of Table S6 (and its link to equation 1) is not straightforward and I am still not convinced by the second part of your previous answer

"However, for the majority of our data ($dit < 3\text{km.}$), we still get that the impact of the context is positive and significant, as in the result in the main paper."

nor by the second part of the sentence below

"Finally, as expected, we find that distance between context and lunch choice modulates this effect, although, for the majority of our data, we still get that the impact of context is positive and significant."

First, it is not clear to me why the positive and significant impact of the context when the distance between context and lunch choice is low (because the majority of the data exhibits low distance between context and lunch choice) has anything to do with the impact of the distance on your results (assuming that all classes of distances are well represented). Second, I am not a specialist of logit model with interaction but the ratio $3.335/2.091=1.6$ seems to indicate a non negligible impact of the distance on the results (even when $dit < 3\text{ km}$) that could even become negative at some point. I may be wrong but to avoid any misunderstanding I would like to see plotted the evolution of the odd-ratio corresponding to an increase of 10% in FFO visits as a function of the distance between context and lunch choice along with the distribution of the distance between context and lunch choice. This would give the reader a clear and unambiguous notion of the impact of the distance between context and lunch choice on the results and on conclusions of this paper.

Response to the reviewers

Once again, we thank the reviewers for their supportive, critical, and extensive assessment of our work “You are where you eat: Effect of mobile food environments on fast food visits,” submitted for consideration for publication in Nature Communications (reference #NCOMMS-22-32906A-Z).

In the following, we address their new concerns point by point. We also specify the changes made to the main paper and the Supplementary Material to address them.

Reviewer 4

Reviewer Point P 4.1 — First, I would like to apologize to the authors, I was focusing too much on the Supplementary Information during the last round of review and missed some information already present in the revised version of the main paper.

Reply: We thank the reviewer again for her/his thorough revision of our manuscript.

Reviewer Point P 4.2 — Regarding my first concern, there is no discussion about the link between a “stays” and an eventual purchase in a POI. This is not because there is a stop in your trajectory near a place that you can guarantee that an activity has been achieved in this place. This is a general issue and there is not much that you can do about it here but I would be grateful if you could add a sentence about that in the “Limitations” paragraph.

Reply: Thank you for this comment. Indeed there is no guarantee that a visit to a POI implies a purchase in that POI. We have changed the “Limitations” paragraph to: “On the other hand, although it is likely that visits to FO that last more than 5 minutes lead to an eventual purchase, which is supported by the strong association between observed visits to a particular type of food outlet (fast food) and intake of food of that type, there is no complete guarantee. Our results, therefore, serve as a proxy and are a lower bound for the potential FF intake.”

Reviewer Point P 4.3 — I will be less accommodating regarding my second concern. I still believe that the distance between context and lunch choice is really important here and I am not satisfied by the authors’ answer. The interpretation of Table S6 (and its link to Equation 1) is not straightforward and I am still not convinced by the second part of your previous answer:

“However, for the majority of our data ($d_{it} < 3km.$), we still get that the impact of the context is positive and significant, as in the result in the main paper.”

nor by the second part of the sentence below

“Finally, as expected, we find that distance between context and lunch choice modulates this effect, although, for the majority of our data, we still get that the impact of context is positive and significant.”

First, it is not clear to me why the positive and significant impact of the context when the distance between context and lunch choice is low (because the majority of the data exhibits low distance between context and lunch choice) has anything to do with the impact of the distance on your results (assuming that all classes of distances are well represented).

Reply: We thank the reviewer again for this comment. The reviewer is completely right that the effect of distance in general (assuming all distances are equally probable) and the fact that for most of our data distances are small are two different things.

To clarify our point, let us explain the results of Table S6. The model in that table is the traditional logistic regression model with interactions between variables. Those models are well-known in the literature [1]. As a review, interaction terms enable the modeler to examine whether the relationship between the target and the independent variable changes depending on the value of another independent variable. In our case, and as suggested by the reviewer, we implemented this well-known statistical tool to investigate how the relationship between the context c_{it} and the lunch action y_{it} changes with distance. The model can be written like this

$$y_{it} \sim \text{logit}^{-1}[\beta_0 + \alpha_i + \delta_t + d_{it} \times \beta\phi(c_{it})] \quad (1)$$

where the \times operator means typically that the regression also includes a term for the modulating variable. Thus, strictly speaking, the model is:

$$y_{it} \sim \text{logit}^{-1}[\beta_0 + \alpha_i + \delta_t + \beta_d d_{it} + \beta_{dc} d_{it} \phi(c_{it}) + \beta_c \phi(c_{it})] \quad (2)$$

Note that this model can be written like

$$y_{it} \sim \text{logit}^{-1}[\beta_0 + \alpha_i + \delta_t + \beta_d d_{it} + \hat{\beta}(d_{it}) \phi(c_{it})] \quad (3)$$

where the “effective” relationship between y_{it} and c_{it} is a function of the distance $\hat{\beta}(d_{it}) = \beta_c + \beta_{dc} d_{it}$.

Thus, if β_{dc} is statistically significant, there is a modulating effect of the distance in the relationship between the context and the lunch action. Results of Table S6 indicate that this is our case, with $\beta_{dc} = -2.091^{***}(0.0329)$ and thus the effective log-odds depend on the distance as $\hat{\beta}(d_{it}) \simeq 3.335 - 2.091 d_{it}$. The effective log-odds can even turn negative at very large distances $d_{it} > \exp(3.335/2.091) \simeq 4.93km$. (note that distance in the models is actual log-distance).

Thus our results in Table S6 indeed indicate, as the reviewer suggests, that distance is important and modulates the effect of the context in the lunch action. We don’t minimize this fact in our manuscript and, as we said in the previous revision round, we commented on this effect of the distance on our results on the main text.

The reviewer is right in that this fact is different from our second point. That, given that for most of our data $d_{it} < 3km$. then $\hat{\beta}$ is positive and significant for most actions and most people in our dataset. That is, for most actions in our dataset our results are as we get for the general model: more FFO in the context increases the probability that lunch action is a visit to an FFO. Only for a small fraction of our data, the effect of the context is different $\hat{\beta}(d_{it}) < 0$, which might be because people that travel such large distances between context and action go to places that are very different from the context before lunch, given the low correlation between the context environments. In our data, we found that lunchtime visits happen far from context (see new Figure S6) only for less than 15% of the actions. We could have also discarded those actions in the first

place, given the large distance between the context and the action, but we opted to include them for the sake of generality.

However, we believe it is important to show the reader that these two points together are important to understand the effect of context on the lunch action in our data. By definition, our action happens close in time to the context, and thus the distance between context and lunch is constrained to be not very large.

Changes to the text: We have expanded Supplementary Note 11 and Figure S6 to explain the interpretation of the interaction terms in our logistic regression and to comment on the importance of the distance in our results and the main conclusion in the paper.

Reviewer Point P 4.4 — Second, I am not a specialist of logit model with interaction but the ratio $3.335/2.091=1.6$ seems to indicate a non-negligible impact of the distance on the results (even when $d_{it} < 3km.$) that could even become negative at some point. I may be wrong but to avoid any misunderstanding I would like to see plotted the evolution of the odd-ratio corresponding to an increase of 10% in FFO visits as a function of the distance between context and lunch choice along with the distribution of the distance between context and lunch choice. This would give the reader a clear and unambiguous notion of the impact of the distance between context and lunch choice on the results and on conclusions of this paper.

Reply: We refer to our previous answer: distance is important, and modulates the log-odds of the impact of the context on the action as $\hat{\beta}(d_{it})$. Thus the plot that the reviewer asks for is the function $\hat{\beta}(d_{it})$ whose coefficients can be easily obtained from Table S6. We think Table S6 shows unambiguously that distance modulates the effect of the context on the lunch action. In any case, we have added the plot suggested to the reviewer in Supplementary Note 11, Figure S6, where we show the change in the odds (in %) of an increase of 10% of FFO places in the context. From the table it is easy to read that the odds change as $(e^{\hat{\beta}(d_{it}) \times 0.1} - 1) * 100$. As we can see the change in the odds of adding more FFO places to the context is positive and can be very large for most of our data.

The study that the reviewer suggests (running a logistic regression for actions only at a fixed value of d_{it}) is ill-defined because, given the nature of the logistic regression, it will only calculate the effect of the context for individuals that go at the same time to FFO and non-FFO at that particular distance. Thus, this will bias the sample to a very special group of individuals and decrease the statistical significance of the results. Furthermore, it will get a different α_i for each individual at each distance. Finally, since the outcome variable y_{it} depends on the distance, it will bias our results to particular actions. Using interaction terms, we have more statistical power, and the model better reflects the whole behavior of people that can be different close to the context or beyond it.

Changes to the text: We have expanded Supplementary Note 11 and Figure S6 to explain the interpretation of the interaction terms in our logistic regression and to comment on the importance of the distance in our results and the main conclusion in the paper. As suggested by the reviewer, we have added new panels to Figure S6 with the distribution of distance d_{it} between context and action and the change in odds as a function of the distance.

References

- [1] Andrew F Hayes and Jörg Matthes. Computational procedures for probing interactions in OLS and logistic regression: SPSS and SAS implementations. *Behavior research methods*, 41(3):924–936, 2009.

REVIEWERS' COMMENTS

Reviewer #4 (Remarks to the Author):

The authors have addressed all my concerns. Thanks for your patience.

Note that $\log(\text{dit})$ could be used instead of dit in Equation 3 and Table S6 to avoid any confusion.

Response to the reviewers

Once again, we thank the reviewers for their supportive, critical, and extensive assessment of our work “You are where you eat: Effect of mobile food environments on fast food visits,” submitted for consideration for publication in Nature Communications (reference #NCOMMS-22-32906A-Z).

In the following, we address the only concern by reviewer #4

Reviewer 4

Reviewer Point P 4.1 — The authors have addressed all my concerns. Thanks for your patience.

Reply: We thank the reviewer again for her/his thorough revision of our manuscript.

Reviewer Point P 4.2 — Note that $\log(dit)$ could be used instead of dit in Equation 3 and Table S6 to avoid any confusion.

Reply: We have made those changes in Equation 3 and Table S6 from d_{it} to $\log d_{it}$

References